# Volume electron microscopy reveals unique laminar synaptic characteristics in the human entorhinal cortex

Sergio Plaza-Alonso[1,2,3], Nicolas Cano-Astorga[1,2,3], Javier DeFelipe[1,2,3], Lidia Alonso-Nanclares[1,2,3]*

[1]Laboratorio Cajal de Circuitos Corticales, Centro de Tecnología Biomédica, Universidad Politécnica de Madrid, Madrid, Spain; [2]Instituto Cajal, Consejo Superior de Investigaciones Científicas (CSIC), Madrid, Spain; [3]Centro de Investigación Biomédica en Red sobre Enfermedades Neurodegenerativas (CIBERNED), ISCIII, Madrid, Spain

**Abstract** The entorhinal cortex (EC) plays a pivotal role in memory function and spatial navigation, connecting the hippocampus with the neocortex. The EC integrates a wide range of cortical and subcortical inputs, but its synaptic organization in the human brain is largely unknown. We used volume electron microscopy to perform a 3D analysis of the microanatomical features of synapses in all layers of the medial EC (MEC) from the human brain. Using this technology, 12,974 synapses were fully 3D reconstructed at the ultrastructural level. The MEC presented a distinct set of synaptic features, differentiating this region from other human cortical areas. Furthermore, ultrastructural synaptic characteristics within the MEC was predominantly similar, although layers I and VI exhibited several synaptic characteristics that were distinct from other layers. The present study constitutes an extensive description of the synaptic characteristics of the neuropil of all layers of the EC, a crucial step to better understand the connectivity of this cortical region, in both health and disease.

*For correspondence:
aidil@cajal.csic.es

**Competing interest:** The authors declare that no competing interests exist.

## Editor's evaluation

This study presents a useful examination of dense neuroanatomy in human postmortem medial entorhinal cortex, using a large number of small electron microscopy image volumes sampled from multiple cortical layers and individuals. The authors use solid experimental and annotation techniques, demonstrating the suitability of postmortem tissue reconstructions for analysis and presenting careful, detailed measurements of synapse properties and overall tissue composition in this brain region. This work would be of interest for studies of cellular neuroanatomy or brain network organization.

## Introduction

The formation and consolidation of declarative memories require the interaction of rich networks that encompass multiple brain areas. Among them, the entorhinal–hippocampal connectivity is a key element (*Buzsáki and Moser, 2013*). In primates, the entorhinal cortex (EC) is a brain region located in the anterior part of the medial temporal lobe, close to the hippocampal formation and the perirhinal (PRC) and parahippocampal (PHC) cortices (*Insausti et al., 2017*). The EC acts as a major sensory interface, mediating the flow of information between the hippocampus and the neocortex (*Eichenbaum et al., 2012*; *Insausti and Amaral, 2008*; *Schultz et al., 2015*). In addition, several studies have

linked major neurodegenerative disorders, such as Alzheimer's disease, to alterations in the EC (*Braak and Braak, 1992*).

The EC is a complex brain region in terms of cytoarchitecture and connectivity. In human and non-human primates, it has been traditionally divided into several subdivisions, based on cytoarchitectonic features (*Insausti et al., 2017*). An alternative organization, which follows the connectivity patterns between the EC, the hippocampus, and parahippocampal regions, divides this area into two functionally defined parts: lateral and medial EC (LEC and MEC, respectively; *Maass et al., 2015*). Briefly, LEC is highly connected to the PRC, providing the hippocampus with information about the content of an experience. As for MEC, it receives projections from the PHC (or the post-rhinal cortex in rodents) and provides the hippocampus with the spatial context of an experience (*Knierim et al., 2014*; *Reagh and Yassa, 2014*). This connectivity-based division was first described in rodents and later in both human and non-human primates (*Maass et al., 2015*). The primate LEC comprises the rostrolateral region of the EC, whereas MEC extends through the caudomedial axis (*Maass et al., 2015*; *Witter and Amaral, 2021*).

The connectivity between EC represents the major cortical afferent source of the hippocampus and has been studied in great detail (*Insausti and Amaral, 2008*; *Witter and Amaral, 1991*). EC upper layers (II and III) give rise to the perforant pathway (*Insausti and Amaral, 2008*; *Insausti and Amaral, 2012*; *Nilssen et al., 2019*). In general, the classic trisynaptic circuit EC layer II → DG → CA3 → CA1 seems to be related to the acquisition of new memories, whereas the direct connection of EC layer III neurons with CA1 (and subiculum; S) (monosynaptic pathway) is thought to contribute to the strength of previously established memories (*Cohen and Squire, 1980*). Deep layers (V and VI) of the EC represent the main target of return connections from the hippocampus, which arise mainly from the CA1 and S (*Insausti and Amaral, 2008*; *Saunders and Rosene, 1988*). These deeply located neurons of the EC project to subcortical and cortical regions such as the amygdala, nucleus accumbens, the temporal pole and the superior temporal sulcus, as well as to frontal areas (*Insausti and Amaral, 2008*; *Muñoz and Insausti, 2005*; *Ohara et al., 2021*). Recent studies have proposed that these deep layers of the EC not only represent simply a relay station between the hippocampus and other telencephalic regions, but also act as a crucial hub for information processing in the EC—hippocampal connectivity. For example, it has been proposed that deep layers of the EC integrate diverse inputs, on a multistep circuit between the hippocampus, the cortex and the superficial EC layers, which may be essential for episodic and spatial memory (reviewed in *Gerlei et al., 2021*).

Most of the connectivity-based studies mentioned have been performed in experimental animals, mainly in rodents and monkeys. Although connectivity appears to be similar in all species studied, there are also species-specific variations. For this reason, we can only assume that the general pattern of connections of the human hippocampal formation is similar to that described in experimental animals, particularly non-human primates, but it is possible that there are significant differences. For example, in the monkey, only the most rostral part of the hippocampus is connected by commissural fibers from the subiculum to the contralateral EC (*Insausti and Amaral, 2012*). In humans, commissural connections seem to be even more reduced (*Insausti and Amaral, 2012*). Recently, *Ben-Simon et al., 2022* reported that excitatory neurons in layer VIb of the mouse EC project to all subregions of the hippocampal formation and receive input from the CA1, thalamus and claustrum.

Connectivity in the brain is mostly based on point-to-point chemical synapses (*DeFelipe, 2015*). Therefore, the synaptic properties of a region are a crucial aspect to be described, both in terms of connectivity and functionality. We can distinguish two major goals in studying synapses. First, studying connections between identified neurons, which consists of identifying the specific presynaptic and postsynaptic neurons involved in each synapse. Second, studying synaptic features in general, which includes quantifying synaptic density, identifying different types and sizes of synapses, and determining their postsynaptic targets (dendrites, somas, or other structures). Assessing the synaptic density in a particular brain region is crucial for understanding both connectivity and functionality. In particular, both the synaptic density (number of synapses per volume) and excitatory/inhibitory proportions are meaningful parameters to describe the synaptic connections of a particular brain region, which can be considered useful for understanding the synaptic circuits of any brain region.

In the human EC, synaptic organization remains largely unknown. There are several approaches available to study synaptic connectivity at the ultrastructural level in the mammalian brain, with the gold standard being volume electron microscopy. Some scientists consider that the best strategy is to

study a relatively large volume of tissue in a given region using serial block-face electron microscopy to obtain saturated or dense reconstructions, even if only one individual can be examined using this approach due to the technical difficulties associated with this technology (e.g., *Karimi et al., 2020*; *Kasthuri et al., 2015*; *Motta et al., 2019*; *Winding et al., 2023*; *Shapson-Coe et al., 2024*). However, we believe that volume electron microscopy can be exploited much more efficaciously if multiple samples of smaller volumes of tissue of the brain region of interest are examined in several individuals using focused ion beam (FIB)/SEM. Multiple sampling using volume electron microscopy is especially important when examining human brain tissue, not only because of the large size of the brain and extension of particular brain regions, but also because interindividual variability is clearly more pronounced than in animal models. To illustrate the magnitude of the technical challenges involved, it is useful to consider the reconstruction of a single fragment of the human temporal cortex from layer I to layer VI, with dimensions of only 1.7 mm × 2.1 mm × 28.3 microns (0.1 mm$^3$), which represents 0.41 petabytes of data (*Loomba et al., 2022*). If the reconstructed volume is increased to 1 mm$^3$ (3 mm × 2 mm × 170 μm), the data expand to 1.4 petabytes (*Shapson-Coe et al., 2024*). Nevertheless, the volume of the EC is relatively large and varies with age, showing differences between the left and right hemispheres ranging approximately from 730 to 1100 mm$^3$, depending on age and hemisphere (*Wang et al., 2019*). Thus, in order to explore the synaptic characteristics present in a given region, our approach is to determine the range of variability by sampling relatively small volumes of the region of interest multiple times and in several individuals. More specifically, we used this technology to obtain data on the density and spatial distribution of different types of synapses (excitatory and inhibitory), as well as to determine the size and shape of the synaptic junctions and the postsynaptic targets of the axon terminals.

Our previous studies of the EC in autopsy tissue have analyzed the neuropil of layers II and III using FIB/SEM (*Domínguez-Álvaro et al., 2021a*). However, in these studies, we did not distinguish between the neuropil of the cell island and transition zones that characterize layer II (*Insausti et al., 2017*; *Kobro-Flatmoen and Witter, 2019*), nor were the rest of the cortical layers and subdivisions analyzed. In the present article, we further examined the ultrastructure of the normal human MEC by analyzing brain tissue obtained at autopsy with FIB/SEM. The purpose was to perform a volume electron microscopy study (3D analysis) in the neuropil of the MEC, including all layers and subdivisions, namely I, II (cell islands and transition zones), III, V (a/b and c), and VI to define the ultrastructural characteristics of synapses. This dataset is critical not only for understanding connectivity, but also from a functional perspective, and it is currently virtually unavailable for the human brain. This kind of analysis has already yielded high-quality data on synaptic ultrastructural properties from other regions of the human cerebral cortex obtained at autopsy (e.g., *Cano-Astorga et al., 2021*; *Domínguez-Álvaro et al., 2021a*; *Montero-Crespo et al., 2020*). Thus, the data presented here aim to represent a step toward a better understanding of the networks and connectivity characteristics of the human cerebral cortex.

## Results

In the present study, we used coronal sections of the MEC at a caudal level—following the subarea delimitation proposed in *Insausti et al., 2017*—to analyze layers I, II, III, IV, V, and VI of the EC. The layers were individually delimited based on cytoarchitectural characteristics, identified by two different staining techniques (i.e., Nissl and toluidine blue on semithin sections), as well as by direct visualization using FIB/SEM. Layer I showed a uniform, thick band with very few neurons present. Layer II presented clear islands occupied by large, stellate cells and modified pyramidal neurons, and transition zones with very few neurons between islands (*Insausti et al., 2017*; *Kobro-Flatmoen and Witter, 2019*). This clear separation enabled us to study both areas separately (referred to here as 'layer II-is' for the area occupied by the islands, and 'layer II-ni' for the neuropil between islands). Layer III comprised a homogeneous population of medium-sized pyramidal neurons, radially organized (*Insausti et al., 2017*). Layer V, in turn, was characterized by the presence of large pyramidal neurons homogeneously distributed throughout the layer (*Insausti et al., 2017*; *Insausti and Amaral, 2012*). This layer is traditionally divided into three subdivisions: layers Va, Vb, and Vc. However, at the level analyzed in the present study, sublayers Va and Vb are often indistinguishable (*Insausti et al., 2017*). Therefore, here we considered both as a single layer (for the sake of clarity, we will refer to it as Va/b). Layer Vc consisted of a pronounced band formed by a few neurons and a dense plexus of

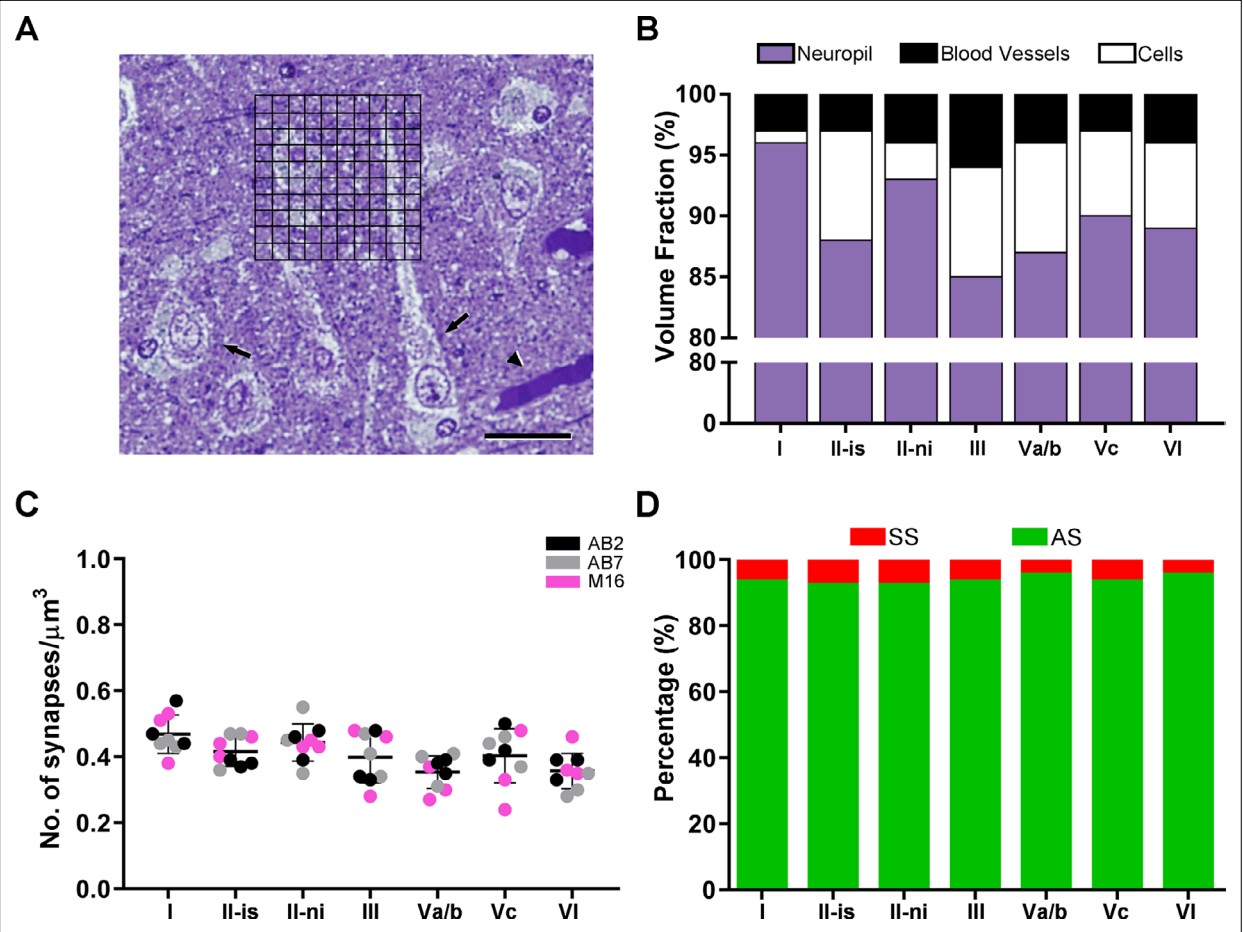

**Figure 1.** Microatatomical analyses of the medial entorhinal cortex (MEC). (**A**) Semithin section, stained with toluidine blue, with a superimposed grid, where points hitting the different cortical elements were counted (grid spacing of 5 μm). Black arrowhead indicates a blood vessel; black arrows indicate cell bodies. (**B**) Graph showing the volume fraction of every cortical element. Values are detailed in *Supplementary file 1a and b*. (**C**) Mean synaptic density (± SD) per layer of the MEC. Each colored dot represents a stack of images from the analyzed cases AB2, AB7, and M16 (see *Supplementary file 1p* for details). No differences in the mean synaptic densities were found between layers (Kruskal–Wallis [KW]; p>0.05). (**D**) Proportion of asymmetric synapses (AS) and symmetric synapses (SS) per layer, expressed as percentages. Scale bar (in **A**): 25 μm.

The online version of this article includes the following figure supplement(s) for figure 1:

**Figure supplement 1.** Spatial distribution analysis of synapses in the MEC.

**Figure supplement 2.** Interindividual variability of synaptic density and intersynaptic distance in the MEC.

fibers. In the analyzed sections, this sublayer became apparent and it could be mistaken for the caudal extension of the lamina dissecans (or layer IV) (*Insausti et al., 2017*; *Insausti and Amaral, 2012*). Finally, layer VI presented a heterogeneous population of neurons, forming a thick layer that exhibits an abrupt border with the underlying white matter (*Insausti et al., 2017*; *Insausti and Amaral, 2012*). Given these evident cytoarchitectural differences, data from all layers included in the present study were displayed and analyzed separately.

## Neuropil occupies most of the grey matter volume

Estimation of the volume fraction (Vv) occupied by different cortical elements (i.e., neuropil, cells bodies—neuronal, glial, and undetermined somata—and blood vessels) was performed in all MEC layers studied, applying the Cavalieri principle (*Gundersen et al., 1988*). The neuropil constituted the main element in all layers, with the minimum proportion found in layer III (86%) and the maximum in layer I (96%). (*Figure 1*; *Supplementary file 1a and b*). Cell bodies occupied a variable range, with the lowest Vv in layer I (1%) and the highest in layers II-is, III, and Va/b (9% in each layer) (*Figure 1*;

**Table 1.** Accumulated data obtained from the ultrastructural analysis of neuropil from layers I, II-is, II-ni, III, Va/b, Vc, and VI of the MEC.

Data in parentheses are not corrected for shrinkage. AS, asymmetric synapses; CF, counting frame; SS, symmetric synapses; SAS, synaptic apposition surface.

| Layer | No. of AS | No. of SS | No. all synapses | % AS (mean) | % SS (mean) | CF volume (µm³) | No. AS /µm³ (mean ± SD) | No. SS /µm³ (mean ± SD) | No. all synapses/ µm³ (mean ± SD) | Area of SAS AS (nm²; mean ± SE) | Area of SAS SS (nm²; mean ± SE) | Intersynaptic distance (nm; mean ± SD) |
|---|---|---|---|---|---|---|---|---|---|---|---|---|
| I | 1265 | 80 | 1345 | 94 | 6 | 2585 (3265) | 0.44 ± 0.06 (0.39 ± 0.05) | 0.03 ± 0.01 (0.03 ± 0.01) | 0.47 ± 0.06 (0.41 ± 0.05) | 105,906 ± 3583 (98,493 ± 3332) | 53,367 ± 2337 (49,631 ± 2173) | 850 ± 64 (824 ± 62) |
| II-is | 1255 | 100 | 1355 | 92.6 | 7.4 | 2949 (3142) | 0.38 ± 0.04 (0.40 ± 0.05) | 0.03 ± 0.01 (0.03 ± 0.01) | 0.42 ± 0.04 (0.43 ± 0.05) | 118,070 ± 6221 (109,805 ± 5786) | 67,492 ± 5135 (62,768 ± 4776) | 876 ± 68 (850 ± 66) |
| II-ni | 1290 | 97 | 1387 | 93.1 | 6.9 | 2820 (3018) | 0.41 ± 0.05 (0.43 ± 0.06) | 0.03 ± 0.01 (0.03 ± 0.01) | 0.44 ± 0.06 (0.46 ± 0.06) | 118,550 ± 5606 (110,251 ± 5214) | 73,141 ± 6195 (68,022 ± 5762) | 845 ± 52 (819 ± 51) |
| III | 1216 | 74 | 1290 | 94.1 | 5.9 | 2901 (3013) | 0.38 ± 0.07 (0.40 ± 0.08) | 0.02 ± 0.01 (0.02 ± 0.01) | 0.40 ± 0.08 (0.43 ± 0.07) | 130,268 ± 5734 (121,150 ± 5333) | 74,248 ± 9112 (69,051 ± 8474) | 882 ± 90 (856 ± 87) |
| Va/b | 1145 | 50 | 1195 | 95.8 | 4.2 | 3017 (3131) | 0.34 ± 0.05 (0.36 ± 0.06) | 0.01 ± 0.005 (0.01 ± 0.005) | 0.35 ± 0.05 (0.38 ± 0.06) | 136,111 ± 5571 (126,583 ± 5181) | 74,698 ± 7914 (69,469 ± 7360) | 899 ± 48 (871 ± 46) |
| Vc | 1257 | 87 | 1344 | 93.5 | 6.5 | 2983 (3118) | 0.38 ± 0.08 (0.40 ± 0.9) | 0.03 ± 0.01 (0.03 ± 0.01) | 0.41 ± 0.08 (0.43 ± 0.09) | 126,298 ± 5071 (117,457 ± 4716) | 87,377 ± 4716 (81,377 ± 4408) | 865 ± 66 (840 ± 64) |
| VI | 1177 | 53 | 1230 | 95.6 | 4.4 | 3112 (3239) | 0.34 ± 0.05 (0.37 ± 0.06) | 0.02 ± 0.01 (0.02 ± 0.01) | 0.36 ± 0.05 (0.38 ± 0.06) | 99,489 ± 6438 (92,525 ± 5987) | 69,365 ± 10,685 (64,509 ± 9937) | 851 ± 77 (826 ± 75) |
| I-VI | 8605 | 541 | 9146 | 94.1 | 5.9 | 20,367 (21,926) | 0.38 ± 0.03 (0.39 ± 0.02) | 0.02 ± 0.01 (0.02 ± 0.01) | 0.41 ± 0.04 (0.42 ± 0.03) | 119,242 ± 2512 (110,895 ± 2336) | 71,384 ± 2806 (66,387 ± 2610) | 867 ± 67 (841 ± 65) |

*Supplementary file 1a and b*). The Vv of blood vessels ranged from 3% in layer I to 6% in layer III— and the rest of the layers presented values within this range (*Figure 1*; *Supplementary file 1a and b*).

## Electron microscopy

A total of 63 stacks of images were obtained in the neuropil of layers I, II, III, Va/b, Vc, and VI using FIB/ SEM. All results presented here were obtained exclusively from the neuropil, which only comprises those synapses located among the cell bodies and blood vessels, thereby excluding perisomatic synapses or those established on the proximal parts of the apical and basal dendrite trunks.

The number of sections per stack ranged from 229 to 319 (*Supplementary file 1c*), which corresponds to a raw volume ranging from 360 to 502 µm³ (mean: 450 µm³). A total of nine stacks of images per layer were obtained to analyze the neuropil (three stacks of images per case studied, in a total volume of 28,476 µm³; *Supplementary file 1c*). Details of the sizes and volumes of each individual stack of images are provided in *Supplementary file 2*.

## No differences in the synaptic density are found between layers

A total of 12,974 synapses from the 63 stacks of images examined were individually identified, and its synaptic junctions were reconstructed in 3D. After discarding incomplete synapses and those touching the exclusion edges of the unbiased counting frame (CF), 8605 synapses were finally analyzed. In layer I, 1345 synapses were analyzed in a total volume of 2585 µm³. A total of 1355 synapses were analyzed in layer II-is, in a total volume of 2949 µm³. For layer II-ni, 1387 synapses were analyzed, in a volume of 2820 µm³. Similarly, in layer III, 1290 synapses were analyzed in a total volume of 2901 µm³. In layer Va/b, 1195 synapses were analyzed in a total volume of 3017 µm³. In the case of the layer Vc samples, a total of 1344 synapses were analyzed in a volume of 2983 µm³. Lastly, 1230 synapses were analyzed in layer VI, in a total volume of 3112 µm³ (*Table 1*; *Supplementary file 1d*).

Synaptic density values were obtained by dividing the total number of synapses included within the unbiased CF by its volume. The mean synaptic density considering all samples and layers was 0.41 synapses/µm³. There were some differences between layers, although none of these were found to be statistically significant (Kruskal–Wallis [KW], p>0.05). Specifically, layer I had the highest density of synapses (0.47 synapses/µm³). Layers II and III had similar density values (Layer II-is, 0.42 synapses/

$\mu m^3$; layer II-ni, 0.44 synapses/$\mu m^3$; layer III, 0.40 synapses/$\mu m^3$). Layers Va/b and VI, in turn, shared a similar number of synapses per volume unit (0.35 synapses/$\mu m^3$ and 0.36 synapses/$\mu m^3$, respectively), whereas layer Vc had a higher density of 0.41 synapses/$\mu m^3$ (*Figure 1C*, *Table 1*, *Supplementary file 1d*).

## Most synapses are AS, with a similar AS:SS ratio between layers

Asymmetric synapses (AS) are considered to be mostly glutamatergic and excitatory, while symmetric synapses (SS) are mostly GABAergic and inhibitory (*Ascoli et al., 2008*; *DeFelipe and Fariñas, 1992*). Since the synaptic junctions of all synapses analyzed in this study were fully reconstructed in 3D, we classified each of them as AS or SS based on their postsynaptic density (PSD) thickness, and the proportions of AS and SS were calculated in each layer. The AS:SS ratio ranged from 93:7 in layer II-is to 96:4 in layers Va/b and VI (*Figure 1D*, *Table 1*, *Supplementary file 1d*).

## Synaptic distribution is indistinguishable from a random spatial distribution

To study the spatial distribution of the synapses, the actual position of each of the synapses in each stack of images was compared with a random spatial distribution model complete spatial randomness (CSR). For this, the functions F, G, and K were calculated (see 'Materials and methods') in the 63 stacks included in the present study. In the vast majority of stacks analyzed, the three spatial statistical functions resembled the theoretical curve that simulates the random spatial distribution pattern, indicating that synapse distributions could not be distinguished from the random spatial distribution model in all layers (*Figure 1—figure supplement 1*). However, some of the stacks (12 out of 63) showed a tendency toward either regular or clustered grouping, as manifested by one or more statistical functions.

Additionally, we estimated the distance of each synapse to its nearest synapse (i.e., intersynaptic distance). The mean synaptic distance to the nearest neighbor, considering all layers, was 867 nm. The mean data per layer is shown in *Table 1* and *Figure 1—figure supplement 1*. No differences were found between the layers (KW, p>0.05; *Table 1*; *Supplementary file 1d*).

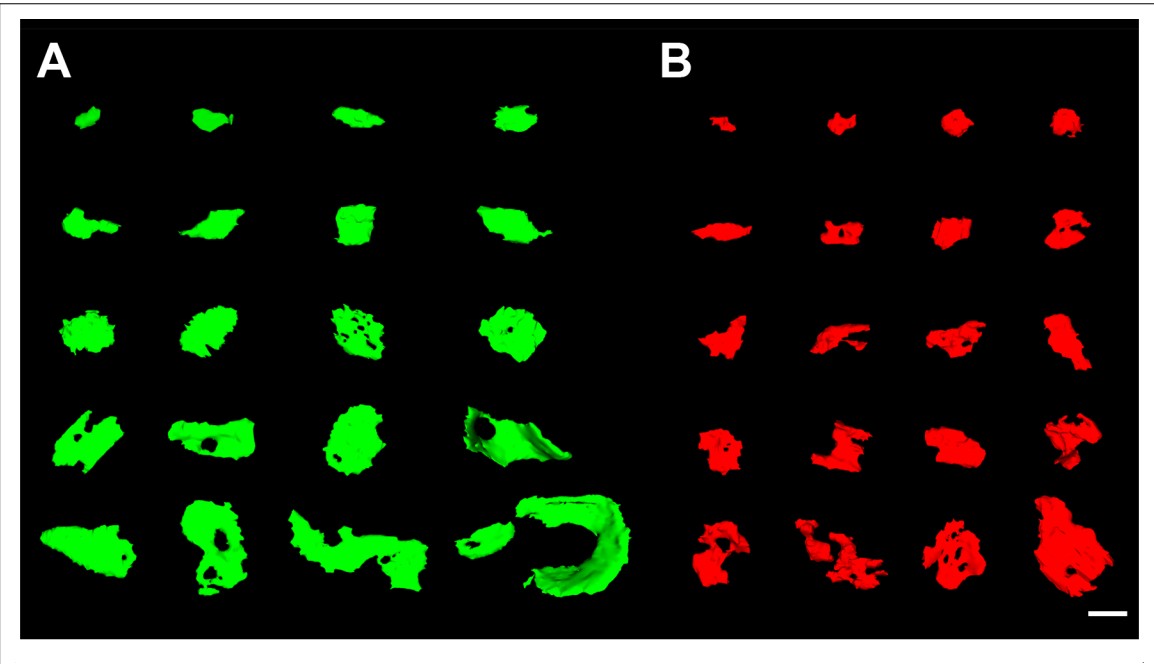

**Figure 2.** 3D representative sample of synaptic apposition surfaces of asymmetric synapses and symmetric synapses in the MEC. (**A**) SAS of asymmetric (green) synapses was distributed into 20 bins of equal size. An example synapse of each bin is illustrated here. (**B**) SAS of symmetric (red) synapses, distributed and represented as in (**A**). Scale bar (in **B**): 240 nm in (**A, B**).

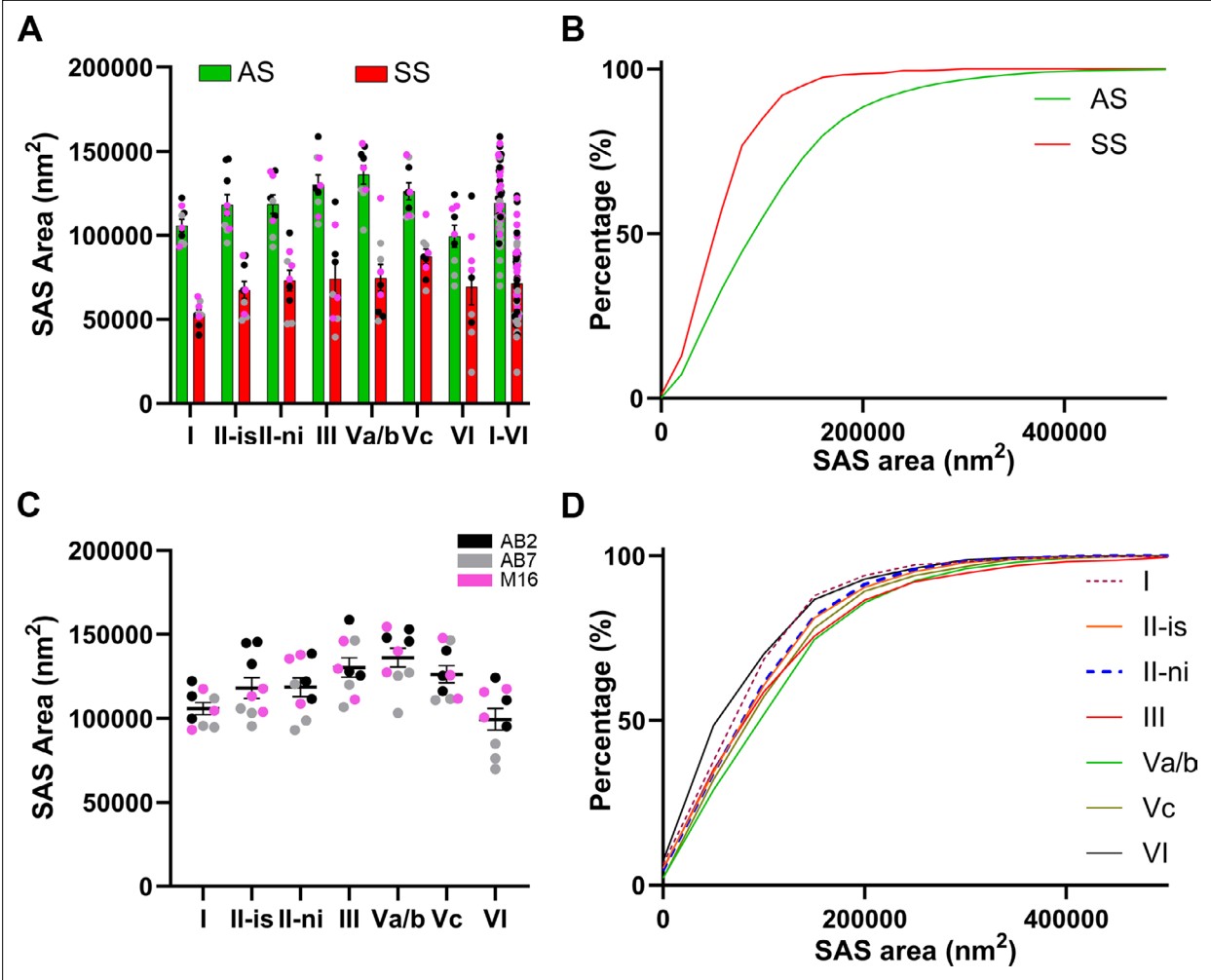

**Figure 3.** Analysis of the synaptic apposition surface (SAS) area of asymmetric synapses (AS) and symmetric synapses (SS) in the MEC. (**A**) Plots of the mean SAS area (± SE) per synaptic type show larger synaptic sizes of AS (green) compared to SS (red) in all EC layers studied (Mann–Whitney [MW], p<0.0001 in layers I, II-is, II-ni, III; p<0.01 in layer Va/b and Vc; p<0.05 in layer VI; p<0.0001 considering all layers). (**B**) Frequency distribution graph of the SAS area illustrating that small SS (red) were more frequent than small AS (green) in all layers (Kruskal–Wallis [KW], p<0.0001). (**C**) Plot of the mean SAS area of AS (± SE) in all layers, with the smallest values in layer VI (Dunn's test, p<0.05). Each colored dot represents a stack of images from the analyzed cases AB2, AB7, and M16 (see *Supplementary file 1p* for details). (**D**) Frequency distribution plot of SAS area per layer, showing that smaller sizes were more frequent in layer VI (Kolmogorov–Smirnov [KS], p<0.0001). Layer II-is had the highest interindividual variability for SAS area of AS, and statistical differences between cases were found (see detailed analysis of the variability in *Figure 3—figure supplement 1* and *Supplementary file 3*).

The online version of this article includes the following figure supplement(s) for figure 3:

**Figure supplement 1.** Interindividual variability of synaptic apposition surface (SAS) area of asymmetric synapses (AS) and symmetric synapses (SS) in the MEC.

## AS are larger than SS

The study of the synaptic size was performed by analyzing the area of the synaptic apposition surface (SAS) of each synapse identified and 3D reconstructed in all stacks of images (*Figure 2*). Considering all layers, the mean SAS areas of AS and SS were 119,242 nm² and 71,384 nm², respectively (*Table 1*; *Supplementary file 1d*). Data regarding AS and SS area per layer are shown in *Table 1* and *Supplementary file 1d*. Regardless of the layer analyzed, AS presented significantly larger SAS areas than SS (Mann–Whitney [MW], p<0.0001 in layers I, II-is, II-ni, III; p<0.01 in layer Va/b and Vc; p<0.05 in layer VI; p<0.0001 considering all layers; *Figure 3*).

The analyses of the mean SAS area of AS were also performed to compare the analyzed cortical layers (*Table 1*; *Supplementary file 1d*). The SAS area of AS showed differences between layers (KW, p<0.0001). We did observe that layer VI had smaller AS (99,489 nm²) than layer III and Vab, which

showed the largest AS (130,268 nm$^2$ and 136,111 nm$^2$; Dunn's test, p<0.05; *Figure 3*). Additionally, layer Va/b showed a larger mean SAS than layer I (105,906 mm$^2$; Dunn's test, p<0.01). Moreover, these differences were also found when comparing the probability density functions between these two layers, indicating that smaller SAS areas of AS were more frequent in layer VI (Kolmogorov–Smirnov [KS], p<0.0001; *Figure 3*). The size of SS was also compared between MEC layers. Layer I had the lowest mean values (53,367 nm$^2$), whereas layer Vc showed the largest mean sizes (87,377 nm$^2$; Dunn's test; p<0.05; *Table 1*).

In order to study the frequency distribution of the SAS area between AS and SS for each individual layer, we performed probabilistic density functions, revealing that smaller synapses were more frequent in SS than in AS, in all layers studied (KS, p<0.0001 in layers I, II-is, II-ni, III, Va/b and Vc; p<0.05 in layer VI; p<0.0001 considering all layers; *Figure 3*).

## SAS area fits into a log-normal distribution

The distribution function of SAS area data of AS and SS was also studied in each layer. For this purpose, we carried out goodness-of-fit tests to find the theoretical probability density functions that best fitted the empirical distribution of the SAS area in each layer. We found that both types of synapses (AS and SS) can be fitted to a log-normal probability distribution, although with some variations in the parameters of the functions (μ and σ) between layers (*Supplementary file 1e*). The fit to the theoretical log-normal function was better for the AS, given the smaller number of SS analyzed in the present study. Considering all layers pooled together, the distribution of AS and SS SAS area data also fitted to a theoretical log-normal probability distribution.

## Macular small synapses are the most frequent regardless of the layer

To analyze the shape of the synaptic contacts, we classified each identified synapse into one of four categories: macular (with a flat, disk-shaped PSD), perforated (with one or more holes in the PSD), horseshoe (with an indentation in the perimeter of the PSD), or fragmented (with two or more physically discontinuous PSDs). For the sake of clarity, the last three categories (perforated, horseshoe, and fragmented) were considered complex-shaped synapses. The percentages regarding macular and complex synaptic shapes for AS and SS are shown in *Figure 4* and *Supplementary file 1f* (see also *Supplementary file 1g* for values for each individual case). The majority of both AS and SS presented a macular shape, regardless of the cortical layer. However, we found some significant differences in the distribution of these synaptic shapes between layers. Particularly, macular synapses (both AS and SS) were less frequent in layer VI (78 and 59%, respectively), whereas the highest values were found in layer II-is ($\chi^2$, p<0.0001; *Figure 4—figure supplement 1*).

The proportion of the synaptic type (AS and SS) on macular and complex synaptic shapes was also determined. Considering all layers, 95% of all macular synapses were AS and 5% were SS. In the case of complex synapses, 92% were AS, with the rest being SS. Slight variations in these percentages were found in the different layers (shown in *Figure 4—figure supplement 2*). No significant differences were found compared to the general distribution of AS:SS ($\chi^2$, p>0.0001).

## Macular synapses have the smallest sizes, regardless of the layer

We also determined whether the shape of the synapses was related to their size. For this purpose, the area of the SAS was analyzed for each synaptic shape. Macular AS synapses presented a significantly smaller mean size compared to complex synapses, regardless of the layer (MW; p<0.0001 in all layers; *Figure 4* and *Supplementary file 1h*; see also *Supplementary file 1i* for values for each individual case). Macular SS synapses also presented a smaller mean size compared to complex-shaped synapses, considering all layers (MW, p<0.0001; *Figure 4*). However, these differences were not as evident as for AS, and most layers did not present a marked change in the mean size of the SS according to their shape; only layers I and VI exhibited significant differences (MW, p<0.05 for layer I and p<0.01 for layer VI; *Figure 4—figure supplement 1*). In fact, layer VI presented the smallest macular SS and the biggest complex-shaped SS (44,170 nm$^2$ and 110,547 nm$^2$, respectively; *Figure 4—figure supplement 1*). Again, these results were also evidenced by the frequency distribution analyses of the SAS (KS, p<0.0001, considering all layers).

However, when comparing macular and complex synapses separately between layers, we also found significant differences (KW, p<0.01). Specifically, layer VI presented both the smallest macular

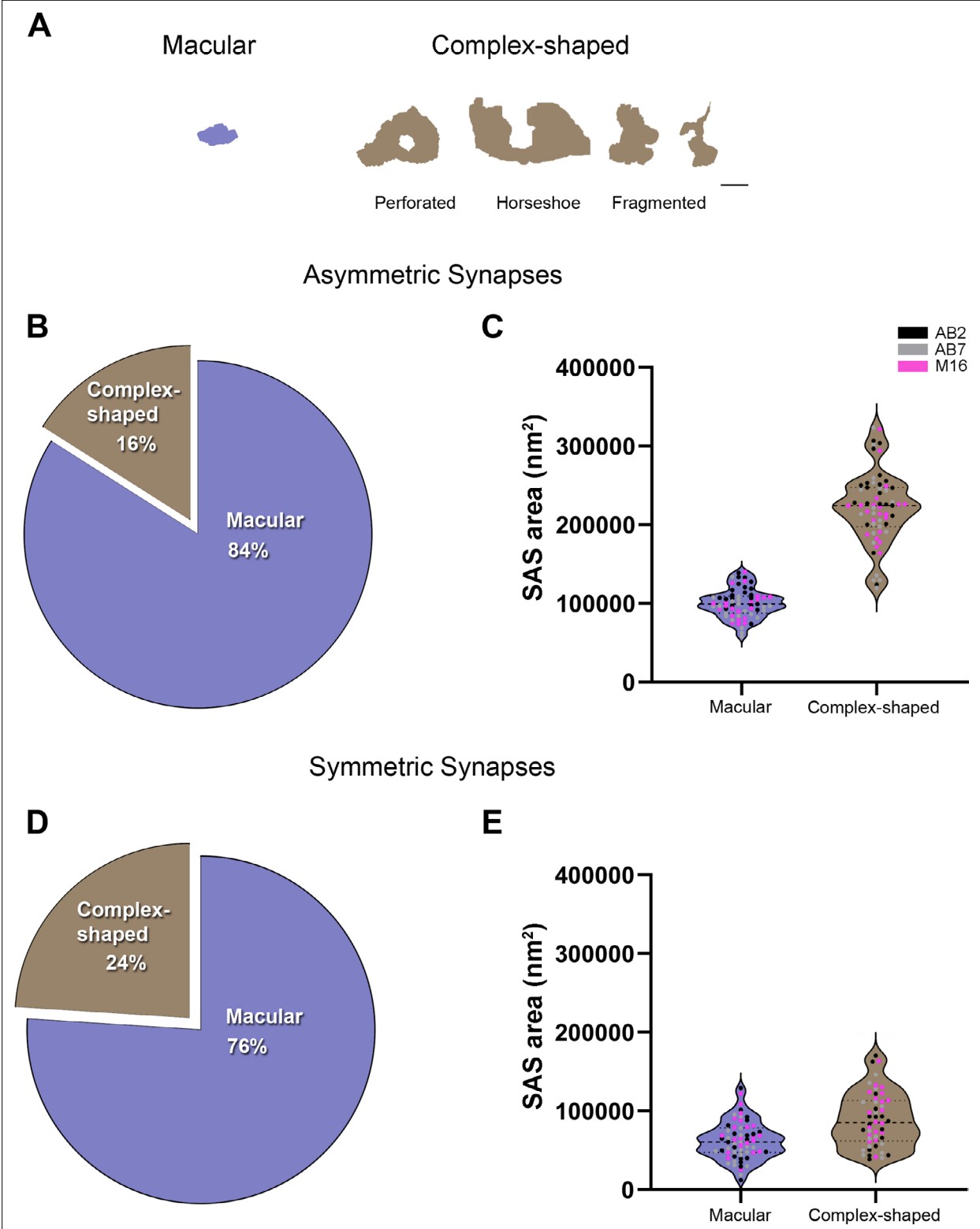

**Figure 4.** Analysis of the synaptic shape of asymmetric synapses (AS) and symmetric synapses (SS) in the MEC. (**A**) Schematic representations of the different types of synapses based on the shape of their synaptic junction: macular, perforated, horseshoe, and fragmented. Perforated, horseshoe, and fragmented were grouped into complex-shaped synapses. Scale bar: 250 nm. (**B**) Pie chart showing the proportion of AS presenting macular and complex-shaped synapses in all layers. Macular synapses represented the most common shape (84%). (**C**) Mean plot of the synaptic apposition surface (SAS) area (± SE) of macular and complex-shaped AS considering all layers. Complex-shaped AS were larger than macular synapses (Mann-Whitney

*Figure 4 continued on next page*

*Figure 4 continued*

[MW], p<0.0001). Each dot represents a stack of images from the analyzed cases AB2, AB7, and M16 (see *Supplementary file 1p* for details). (**D**) Pie chart showing the proportion of SS presenting macular and complex-shaped synapses in all layers. Once again, macular synapses represented the most common shape (76%). (**E**) Mean plot of the SAS area (± SE) of macular and complex-shaped SS considering all layers. Complex-shaped SS were larger than macular synapses (MW, p<0.0001). See *Figure 4—figure supplement 1* for a detailed description of each layer. The statistical interindividual variability test for SAS area of complex AS showed differences between cases in layer VI (see detailed analysis of the variability in *Figure 4—figure supplement 5* and *Supplementary file 3*).

The online version of this article includes the following figure supplement(s) for figure 4:

**Figure supplement 1.** Analysis of asymmetric (AS; **A, B**) and symmetric (SS; **C, D**) synapse shape in the MEC, per layer.

**Figure supplement 2.** Proportion of asymmetric synapses (AS) and symmetric synapses (SS) in macular and complex-shaped synapses.

**Figure supplement 3.** Comparison of the synaptic apposition surface (SAS) of macular (**A, B**) and complex-shaped (**C, D**) asymmetric synapses (AS) between MEC layers.

**Figure supplement 4.** Interindividual variability of synaptic apposition surface (SAS) area of macular asymmetric synapses (AS) and symmetric synapses (SS) in the MEC.

**Figure supplement 5.** Interindividual variability of synaptic apposition surface (SAS) area of complex-shaped asymmetric synapses (AS) and symmetric synapses (SS) in the MEC.

(79,331 nm$^2$) and complex AS (170,959 nm$^2$) of all layers, whereas layer Va/b had the largest SAS for macular synapses (112,898 nm$^2$)—and, in the case of complex-shaped synapses, the largest SAS were found in layer III (244,603 nm$^2$; Dunn's test, p<0.05; *Figure 4—figure supplement 3*). These differences were also shown in the frequency distribution analysis of the SAS for each shape (KS, p<0.0001 in all layers, *Figure 4—figure supplement 3*).

## Most synapses are established on dendritic spines

Postsynaptic targets were identified and classified as dendritic spines (including both complete and incomplete spines, as detailed in 'Material and methods') or dendritic shafts (*Figure 5* displays 3D reconstruction examples of dendritic segments in which synapses were established). In addition, when the postsynaptic element was identified as a dendritic shaft, it was classified as 'spiny' or 'aspiny'. Considering all layers, 59.3% of the total synapses were AS established on dendritic spine heads, 34.3% were AS on dendritic shafts (including spiny and aspiny shafts), 5.1% were SS on dendritic shafts (including spiny and aspiny shafts), 0.7% were SS on dendritic spine heads, 0.5% were AS established on dendritic spine necks, and 0.1% were SS on dendritic spine necks (*Figure 6*, see also *Supplementary file 1j*). When the different layers were compared, layer VI presented the highest proportion of AS established on dendritic spine heads (76.2%), whereas layer Va/b had the lowest value (52.9%, $\chi^2$, p<0.0001; *Figure 6*; *Supplementary file 1j*).

We also analyzed the proportion of AS and SS synapses separately, categorized by their location on dendritic spines (heads or necks) or dendritic shafts. The proportion of AS formed on the different postsynaptic targets greatly resembled the general distribution of synapses established on the postsynaptic targets described above, given the greater number of AS than SS. In all layers of the MEC, AS were mostly established on dendritic spine heads (63%), followed by AS established on dendritic shafts (36.5%; 17.6% on spiny shafts and 18.9% on aspiny shafts) and, lastly, we found a small percentage of AS formed on dendritic spine necks (0.5%; *Supplementary file 1k*). Our analysis revealed significant differences between MEC layers regarding the proportions of AS synapses established on dendritic spine heads. Layer VI exhibited the highest proportion of AS on spine heads (79.6%), whereas layer Va/b had the lowest proportion (54.9%; $\chi^2$, p<0.0001). These findings are illustrated in *Figure 6* and *Supplementary file 1k*.

The distribution of SS on different postsynaptic targets differed significantly from that observed for AS. The majority of SS were found on dendritic shafts (85.7%), with 47.5% on shafts containing dendritic protrusions (spiny shafts) and 38.2% on smooth dendritic shafts (aspiny shafts). A smaller proportion of SS were established on dendritic spine heads (12.1%) and spine necks (2.2%). Due to the low number of SS identified as axospinous in each layer (ranging from 3 in layer Va/b to 19 in layer Vc), statistical comparisons of this proportion between layers were not performed. However, we observed some variations between layers. Layer Va/b had the highest proportion of SS on dendritic shafts

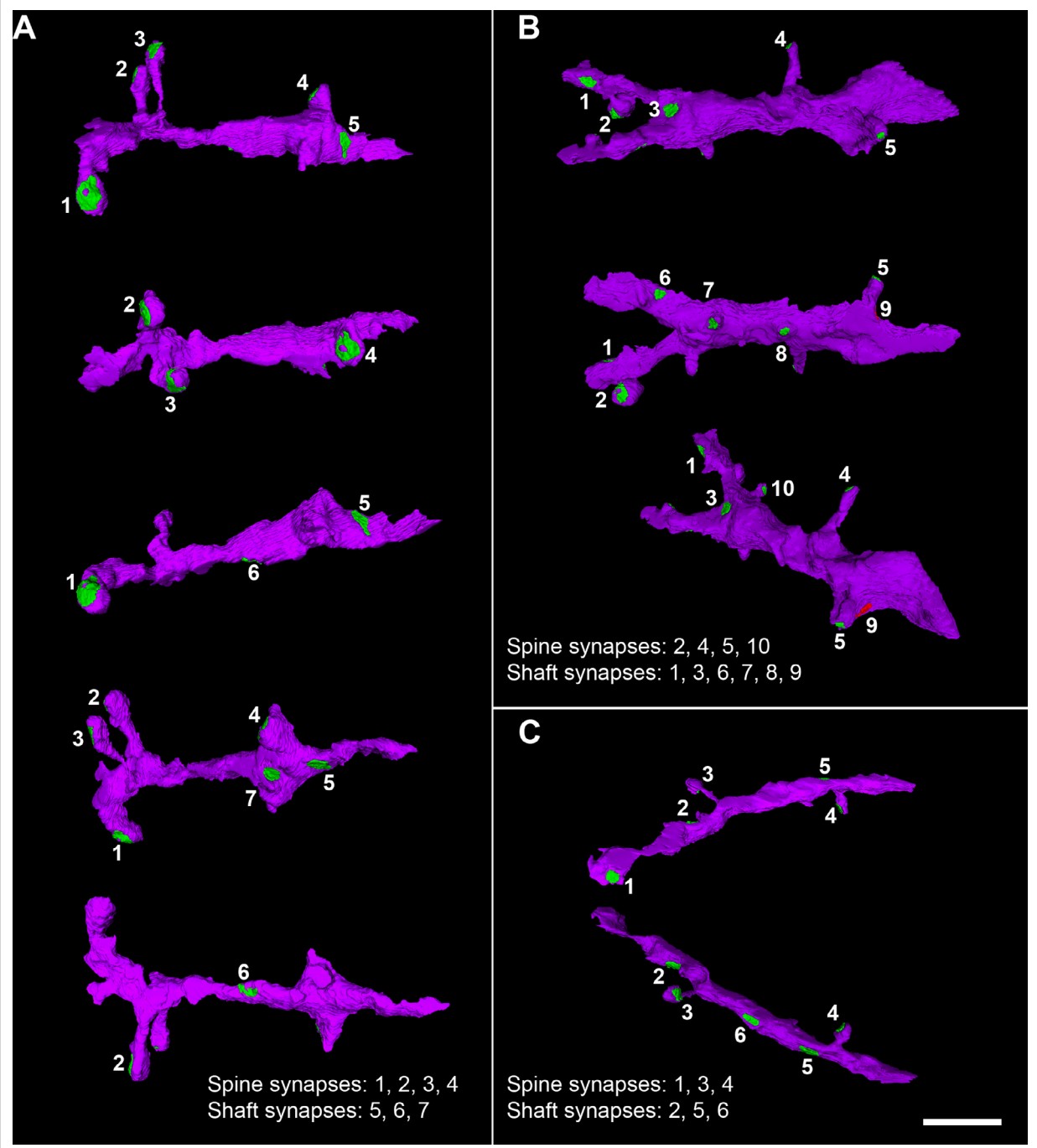

**Figure 5.** Examples of 3D reconstructed dendritic segments (purple) establishing asymmetric (green) and symmetric (red) synapses. Three different dendritic segments (**A–C**) illustrating synaptic junctions at different view angles. Numbers correspond to the same synapse in each dendritic segment. Note the different sizes and shapes of the synaptic junctions on each dendritic segment. The dendritic surface of the shaft presents relatively few synaptic contacts. Scale bar (in **C**): 3.6 µm in (**A–C**). See also *Figure 13*.

(92.3%), whereas layer Vc exhibited the lowest percentage of SS formed on shafts (77%; *Figure 6*; *Supplementary file 1k*).

Considering all types of synapses established on dendritic spines, the proportion of AS:SS was 99:1, whereas the proportion of AS:SS established on dendritic shafts was 87:13. The overall ratio of AS:SS was 96:4, which indicates a 'preference' of AS and SS for a particular postsynaptic destiny: AS are most frequently found on dendritic spine heads, and SS are predominantly established on

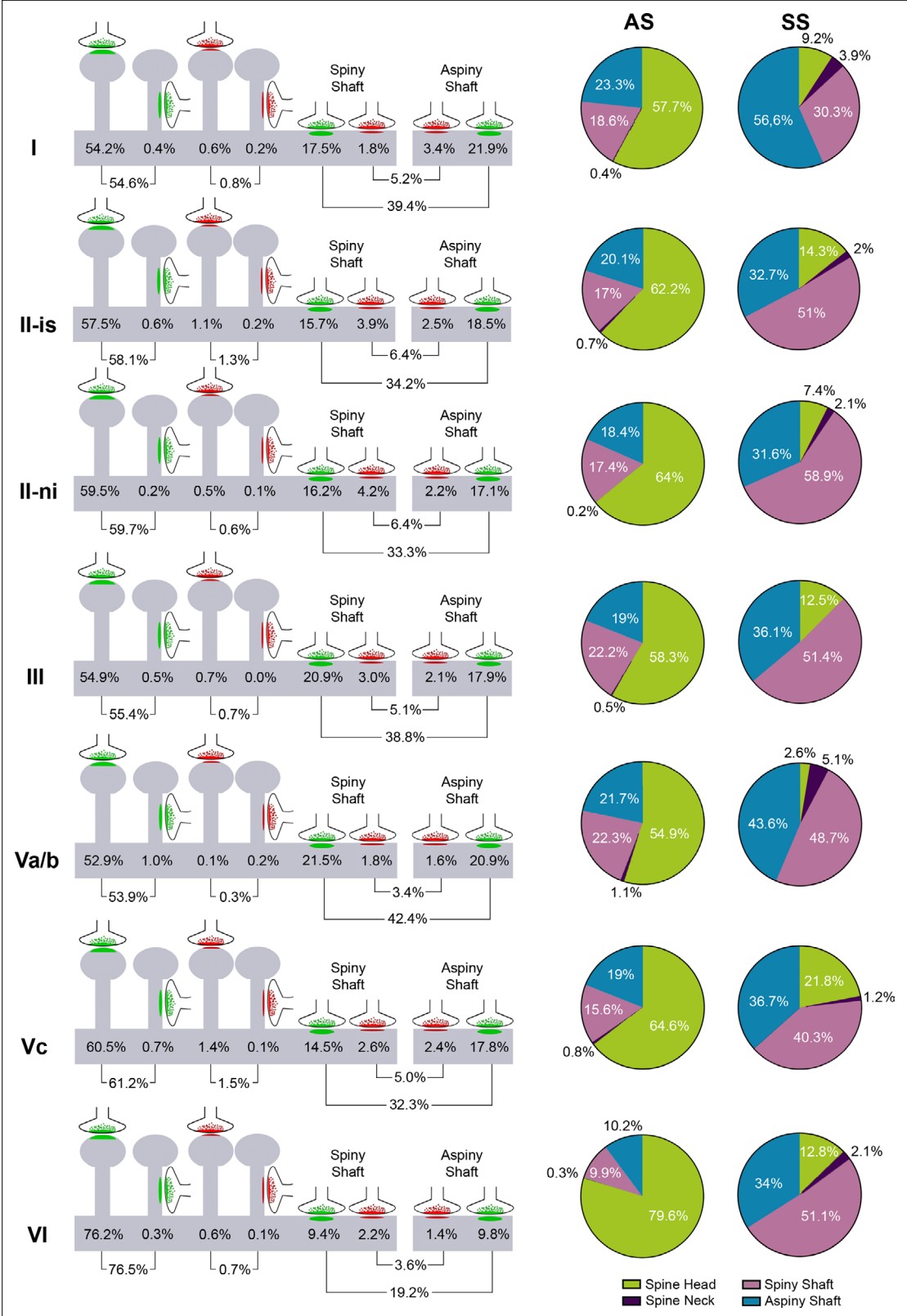

**Figure 6.** Distribution of synapses according to their postsynaptic target in each MEC layer. (Left) Schematic representation of the distribution of synapses according to their postsynaptic element: data refer to the percentages of axospinous (i.e., head and neck of dendritic spines) and axodendritic (spiny and aspiny shafts) asymmetric (green) and symmetric (red) synapses (see *Supplementary file 1j* for detailed information regarding the absolute number of each synaptic type). (Right) Pie charts to illustrate the proportion of asymmetric synapses (AS) (green) and symmetric synapses (SS) (red)

*Figure 6 continued on next page*

*Figure 6 continued*

according to their location as axospinous synapses (i.e., on the head or neck of the spine) or axodendritic synapses (i.e., spiny or aspiny shafts) in each MEC layer (see legend). AS were preferentially formed on dendritic spine heads (63%), although differences were observed between layers ($\chi^2$, p<0.0001). **Supplementary file 1k** shows the percentage and absolute number of each synaptic type.

The online version of this article includes the following figure supplement(s) for figure 6:

**Figure supplement 1.** Proportion of asymmetric synapses (AS) and symmetric synapses (SS) in dendritic spines and shafts.

**Figure supplement 2.** Schematic representation of the proportion of single and multisynaptic spines in all MEC layers.

**Figure supplement 3.** Analysis of synaptic apposition surface (SAS) of asymmetric synapses (AS) for each of the postsynaptic targets in the MEC.

dendritic shafts ($\chi^2$, p<0.0001). These results were consistent across the different layers (**Figure 6— figure supplement 1**).

Additionally, we were able to detect dendritic spines that contained more than one synapse (i.e., multisynaptic spines). The vast majority of dendritic spines (96%) contained only one AS, followed by a small pool of spines that showed different combinations regarding the type (AS or SS), number, and localization (head or neck) of synapses (**Figure 6—figure supplement 2**). There were no significant differences in the proportion of multisynaptic spines between the different layers ($\chi^2$, p>0.0001). Nevertheless, interestingly, the proportion of spines presenting one AS and one SS (0.73%) was higher than the percentage of dendritic spines containing a single SS (0.50%)——and this was consistent in most of the layers (**Figure 6—figure supplement 2**).

## AS established on dendritic spine necks are the smallest

The relationship between the postsynaptic elements of the synapses and their size was also determined. For this purpose, the areas of the SAS—of both AS and SS—were analyzed according to the postsynaptic targets. In order to perform a more accurate analysis of the size of the axospinous synapses, we excluded those synapses established on incomplete spines. Considering all layers, for AS, the mean area of synapses established on dendritic spine heads was similar to those of synapses established on dendritic shafts (127,956 nm$^2$ and 120,978 nm$^2$, respectively). However, AS established on dendritic spine necks were notably smaller considering all layers (66,435 nm$^2$; Dunn's test, p<0.001, **Figure 6—figure supplement 3**; **Supplementary file 1l**; see also **Supplementary file 1m** for values for each individual case). All layers displayed a similar distribution regarding the size of spine heads, necks, and dendritic shafts (**Figure 6—figure supplement 3**; **Supplementary files 1l and m**). When comparing the SAS area of AS on spine heads separately between layers, we found that layer Va/b presented the largest sizes (148,730 nm$^2$) and layer VI the smallest (105,948 nm$^2$) of all layers (Dunn's test, p<0.05). In addition, analysis of the SAS area of AS on shafts showed that layer Va/b had the largest synapses (143,122 nm$^2$) and layer I the smallest (99,960 nm$^2$; Dunn's test, p<0.01). Additionally, the analysis of the size of AS on multisynaptic spines revealed that AS established on AS:SS multisynaptic spines had larger SAS (155,446 nm$^2$) than AS formed on single spines (120,186 nm$^2$; KS, p<0.01).

SS showed similar results. Synapses established on dendritic shafts and spine heads shared a similar value (76,801 nm$^2$ and 70,198 nm$^2$, respectively), whereas SS on spine necks exhibited a smaller SAS (53,608 nm$^2$), considering all layers (**Supplementary files 1l and m**). However, due to the small number of SS established both on dendritic spine heads and necks, we were not able to perform any robust statistical analyses.

## Complex-shaped synapses are more frequently found on dendritic spines

Finally, we examined whether macular and complex synapses have a particular distribution on different postsynaptic targets. The general distribution of AS established on dendritic spine heads and dendritic shafts was 63:37, considering all layers. This proportion ranged from 80:20 in layer VI to 55:45 in layer Va/b. When considering macular AS, 61% of them were established on dendritic spine heads, and 39% were formed on shafts (**Figure 7**). Complex-shaped AS were even more predominantly established on dendritic spine heads (75%, **Figure 7**); in fact, this proportion was significantly different from the general distribution ($\chi^2$, p<0.0001). All layers exhibited similar tendencies (**Figure 7—figure supplements 1 and 2**), although layer VI exhibited both the highest proportion of macular and complex-shaped synapses established on dendritic spines (78% and 85%, respectively).

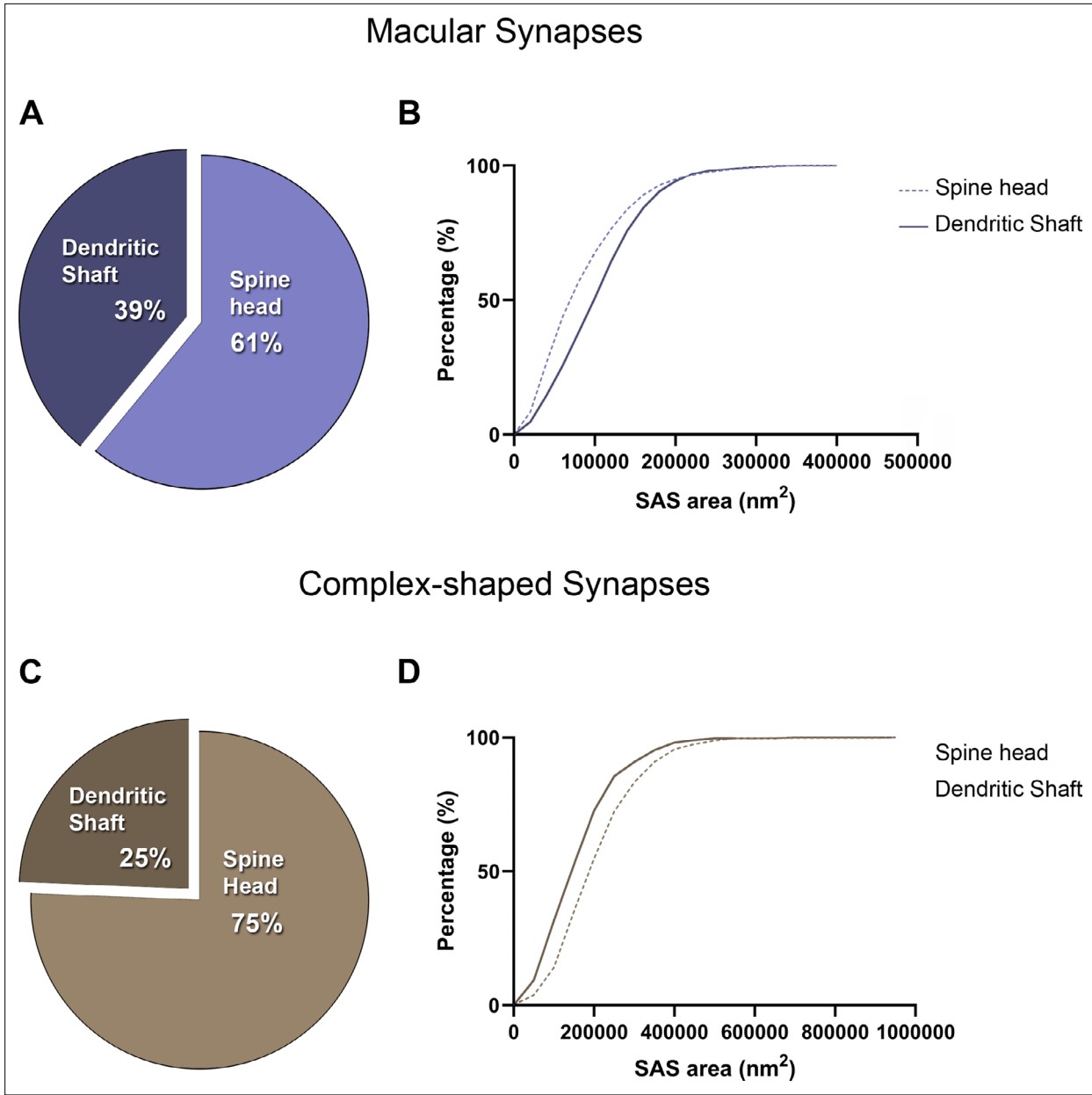

**Figure 7.** Analysis of the distribution of macular and complex-shaped asymmetric synapses (AS) on postsynaptic targets. (**A**) Pie chart illustrating the proportion of macular AS on spine heads and dendritic shafts, considering all layers. This distribution is similar to the general proportion of AS on spine heads and shafts, regardless of the synaptic shape (63:37). (**B**) Frequency distribution plot of the synaptic apposition surface (SAS) of macular AS on spine heads and dendritic shafts, showing that smaller macular AS were more frequent on spine heads (Kolmogorov–Smirnov [KS], p<0.0001). (**C**) Pie chart illustrating the proportion of complex-shaped AS on spine heads and dendritic shafts, considering all layers. In this case, the proportion of complex synapses on spine heads is higher than what would be expected from the general distribution ($\chi^2$, p<0.0001). (**D**) Frequency distribution plot of the SAS of complex-shaped AS on spine heads and dendritic shafts, showing that smaller complex-shaped AS were more frequent on dendritic shafts (KS, p<0.0001). See *Figure 7—figure supplements 1 and 2* for a detailed description of each layer.

The online version of this article includes the following figure supplement(s) for figure 7:

**Figure supplement 1.** Analysis of the distribution of macular asymmetric synapses (AS) on postsynaptic targets, per MEC layer.

**Figure supplement 2.** Analysis of the distribution of complex-shaped asymmetric synapses (AS) on postsynaptic targets, per MEC layer.

To further study this, we carried out frequency distribution analyses of the SAS from the macular and complex synapses established on dendritic spines and dendritic shafts. We found that smaller macular synapses were more frequent on spine heads than on dendritic shafts (*Figure 7*; KS, p<0.0001), whereas smaller complex-shaped synapses were more frequent on dendritic shafts than

**Table 2.** Coefficient of variation (CV) of the analyzed synaptic parameters in each MEC layer between individuals.
CV was calculated by dividing the SD by the mean and multiplying by 100 for each of the synaptic parameters analyzed.
AS, asymmetric synapses; SAS, synaptic apposition surface.

| Layer | Synaptic density | Proportion of AS | AS SAS area | Proportion of macular AS | Proportion of AS on spines |
|-------|------------------|------------------|-------------|--------------------------|----------------------------|
| I | 5.4 | 1.0 | 3.3 | 4.0 | 5.1 |
| II-is | 7.0 | 0.8 | 17.3 | 7.0 | 11.4 |
| II-ni | 1.3 | 3.3 | 10.7 | 6.5 | 10.3 |
| III | 4.3 | 0.4 | 5.4 | 3.9 | 1.1 |
| Va/b | 9.9 | 0.3 | 15.5 | 5.7 | 13.5 |
| Vc | 11.7 | 1.9 | 2.4 | 4.6 | 7.5 |
| VI | 11.7 | 1.7 | 19.6 | 3.8 | 5.0 |

on dendritic spine heads (*Figure 7*; KS, p<0.0001). All layers displayed similar tendencies (*Figure 7— figure supplements 1 and 2*). There were not enough SS established on dendritic spines to perform a robust analysis.

## Synaptic parameters show little variability

Variability between cases was examined by calculating the coefficient of variation (CV, see 'Materials and methods') and by statistically comparing each synaptic parameter in every MEC layer (*Figure 1— figure supplement 2*, *Figure 3—figure supplement 1*, *Figure 4—figure supplements 4 and 5*; p-values are detailed in *Supplementary file 3*).

We observed little variability between cases for any of the synaptic characteristics analyzed (*Table 2*). Overall, the SAS area of the AS exhibited the highest CV values of all the parameters analyzed, ranging from 3.3% in layer I to 19.6% in layer VI (see *Table 2*). Statistical analyses only showed significant differences in two particular comparisons: SAS area of AS in layer II-is (KW, p<0.05) and SAS area of complex AS in layer VI (KW, p<0.01); in both analysis, differences were found between AB2 and AB7 (Dunn's test, p<0.05; *Figure 3—figure supplement 1*, *Figure 4—figure supplement 5*, *Supplementary file 3*).

We also calculated the variability between the stack of images, examining the CV for the same synaptic parameter in each MEC layer. In this case, the highest variability was observed in the synaptic density in layer Vc (20.4%; *Supplementary file 1n*). However, the rest of the layers presented low variability regarding synaptic density. Furthermore, the SAS area of AS presented a relatively higher variability than the rest of the synaptic parameters analyzed, ranging from 10.2% in layer I to 19.4% in layer VI (*Supplementary file 1n*).

## Summary of results

In the present study, we obtained the following findings:

(i) neuropil occupies most of the MEC volume; (ii) no differences in the synaptic density were found between layers; (iii) most synapses were AS; (iv) synaptic distribution was indistinguishable from a random spatial distribution; (v) AS synapses were larger than SS synapses; (vi) similar SAS area of AS was found in all layers, with smaller sizes more frequent, especially in layer VI; (vii) smaller synapses were more frequent in SS than in AS in all layers; (viii) SAS area fitted into a log-normal distribution; (ix) macular synapses were the most frequent synaptic shape, and had the smallest sizes, regardless of the layer; (x) around 60% of the synapses were established on dendritic spines, the vast majority of which were on dendritic spine heads; (xi) AS were preferentially formed on dendritic spines, whereas SS were mostly established on dendritic shafts; (xii) the vast majority of dendritic spines contained only one AS; (xiii) AS established on dendritic spine necks were smaller; (xiv) AS established on AS:SS multisynaptic spines had larger SAS than single AS on spines; and (xv) complex-shaped synapses were more frequently found on dendritic spine heads than on dendritic shafts.

## Discussion

In the present study, we have performed the first detailed description of the synapses of layers I, II (both between and within the cell islands), III, Va/b, Vc, and VI from the human MEC using FIB/SEM technology. The main findings indicate that the MEC exhibits a distinct set of synaptic features that differentiates this region from other human cortical areas. Additionally, our results reveal common patterns in the synaptic structure across mammalian brains, such as the proportion of synaptic types and shapes. Furthermore, ultrastructural synaptic characteristics within the MEC are predominantly similar, although layers I and VI exhibit several synaptic characteristics that are distinct from those in the other layers.

### The synaptic density is similar in all layers and most synapses are excitatory

Synaptic density is considered a meaningful parameter, in terms of connectivity, to describe the synaptic organization of a brain region (*Cano-Astorga et al., 2021*). In the present study, we found that the analyzed MEC layers had similar synaptic density values. Moreover, the average density was 0.41 synapses/µm³, very close to the previously reported values in layers II and III of the intermediate-caudal EC (0.42 synapses/µm³, *Domínguez-Álvaro et al., 2021a*). However, MEC had slightly lower synaptic densities than other cortical regions from human samples studied using identical processing and analyzing methods, such as layer III from Brodmann area 21 (0.49 synapses/µm³, *Cano-Astorga et al., 2023*); layer III from Brodmann area 24 (0.49 synapses/µm³), area 38 (ranging from 0.54 to 0.76 synapses/µm³; *Cano-Astorga et al., 2023*), area 17 and area 3b (0.45 and 0.44 synapses/µm³, respectively, *Cano-Astorga et al., 2024a*), and the CA1 hippocampal field (ranging from 0.45 synapses/µm³ in the stratum oriens to 0.99 synapses/µm³ in the superficial stratum pyramidale; *Montero-Crespo et al., 2020*). These differences were more evident for deep MEC layers (especially Va/b and VI), which had the lowest values (0.35 and 0.36 synapses/µm³, respectively). Indeed, the data presented here highlight the importance of studying the human brain by region and in a layer-specific manner since synaptic values might depend on the specific features of the particular brain region and/or layer. Further analysis of human deep layers from other cortical areas might provide information to determine whether deep layers have a lower synaptic density or whether there is specificity depending on the brain region.

Previous studies, using transmission electron microscopy methods to estimate the synaptic density from layers III and V of the human EC, obtained a density of 0.32 and 0.27 synapses/µm³, respectively (*Scheff et al., 1993*), which is about 30% lower than reported in the present study. This difference in synaptic density may stem from a number of differences between their approach and ours. For example, Scheff et al. treated layer V as a single layer without dividing it into two sublayers as we did. Also, there were differences regarding the age of the human subjects—in our study, the ages of the three individuals examined were 40, 53, and 66 years old, while in the study by Scheff et al., the cases analyzed were older (range: 58–103 years old; mean ± SD, 71.7 ± 11.7 years), with longer postmortem delays (mean ± SD, 8.0 ± 2.8 hr, compared to 4, 2.4, and 3 hr in our cases, *Supplementary file 1p*). Moreover, the method employed to estimate synaptic density differed; Scheff et al. used an estimation of the number of synaptic profiles per unit area using single ultrathin sections. In this approach, known as the size-frequency method, synaptic density is estimated from two-dimensional samples. However, FIB/SEM technology enables us to calculate the actual number of synapses per volume unit, avoiding most of the bias associated with these traditional EM methods (*Cano-Astorga et al., 2021*; *Merchán-Pérez et al., 2009*).

The final number of synapses per cortical layer may be influenced by the proportion of tissue which is occupied by the neuropil. In the present study, the neuropil, where the vast majority of synapses are found (*DeFelipe et al., 1999*), constituted the main element in all of the layers studied (86–96%, see *Figure 1* and *Supplementary file 1a and b*). This result is similar to previous values obtained in several regions of the human neocortex and hippocampus (e.g., *Domínguez-Álvaro et al., 2021a*; *Cano-Astorga et al., 2024a*).

It has been widely reported that the cortical neuropil has a larger proportion of AS than SS, regardless of the cortical region and species analyzed (reviewed in *DeFelipe et al., 2002*, see also *Cano-Astorga et al., 2021*; *Cano-Astorga et al., 2023*; *Cano-Astorga et al., 2024a*). In the present study, the AS:SS ratio was, on average, 94:6, and no differences were found between layers.

However, interpreting the functional relevance of AS:SS is challenging due to variations in cytoarchitecture and connectivity across different cortical regions. Although AS:SS ratios were similar in all MEC layers, we cannot rule out differences in the proportion of AS and SS on the dendritic arbors of particular neurons. For example, previous studies have demonstrated variations in the number of inputs of glutamatergic and GABAergic synapses in different neuronal types (e.g., *Bourne and Harris, 2011*; *DeFelipe, 1997*; *DeFelipe and Fariñas, 1992*; *Hu and Vervaeke, 2018*; *Schubert et al., 2007*). Therefore, examining synaptic inputs for specific cell types is necessary to assess potential differences in the AS:SS ratio in these cells, even though the overall AS:SS ratio may remain constant in the neuropil. Additionally, it is important to mention that our current study exclusively focuses on neuropil synapses, while the analysis of perisomatic innervation remains unexplored. Investigating perisomatic innervation would undoubtedly enhance our comprehension of the synaptic organization of these cortical circuits. The study of the axo-somatic innervation on neuronal somata would require a different technical approach (e.g., see *Ostos et al., 2023*).

## Synaptic distribution is indistinguishable from a random spatial distribution

The present analysis of the synaptic spatial distribution indicated that statistical tests were unable to distinguish synapse distributions from random distributions in the neuropil of most of the FIB/SEM samples.

As discussed in *Merchán-Pérez et al., 2014*, in a random distribution, a synapse could be formed anywhere in space where an axon terminal and a dendritic element may touch, provided this particular spot is not already occupied by a preexisting synapse. However, spatial randomness does not necessarily mean nonspecific connections. In fact, spatial specificity in the neocortex may be scale dependent. At the macroscopic and mesoscopic scales, the mammalian nervous system is highly structured, with connections established in a very specific and ordered way. Even at the microscopic level, different layers of the cortex receive specific inputs.

Previous ultrastructural studies of other human cortical areas, including the CA1 field of the hippocampus, have also reported synapse distributions exhibiting patterns that statistical tests could not distinguish from randomness (*Cano-Astorga et al., 2021*; *Cano-Astorga et al., 2023*; *Cano-Astorga et al., 2024a*; *Domínguez-Álvaro et al., 2021a*; *Montero-Crespo et al., 2020*). These findings suggest that the spatial distribution of synapses in the neuropil may reflect a general tendency toward randomness across the analyzed regions of the human cerebral cortex. This could mean that, as the axon terminals reach their destination, the spatial resolution achieved by them is fine enough to find a specific cortical layer, but not sufficiently fine to form a synapse on a particular target, such as a specific dendritic branch or dendritic spine within a layer.

Nevertheless, a few stacks of images from different layers showed slight tendencies toward either cluster or regular distributions. These 'unexpected' 3D distributions may arise from particular elements of the tissue studied, such as the presence of large trunks of apical dendrites or myelin fibers that could reduce the effective neuropil space where synapses may be established.

## Macular small synapses are the most frequent and have the smallest sizes

Determination of the individual synaptic characteristics (such as synaptic size and shape) provides crucial data on the functionality of the synapses, regarding both synaptic transmission efficiency and dynamics. It has been proposed that synaptic size correlates with release probability, synaptic strength, efficacy, and plasticity (see *Chindemi et al., 2022* and references therein). It has been demonstrated that synaptic size correlates with the number of receptors in the PSD; larger PSDs have more receptors (as reviewed in *Lüscher and Malenka, 2012*; *Lüscher et al., 2000*; *Magee and Grienberger, 2020*; *Sumi and Harada, 2020*; *Toni et al., 2001*). In the present study, MEC layers presented an average SAS area of 119,242 $nm^2$ for AS and 71,384 $nm^2$ for SS. In the MEC layers included in the present study, the analysis of SAS area also revealed that AS were larger than SS, as previously reported in other human cortical regions and layers (e.g., *Cano-Astorga et al., 2021*; *Cano-Astorga et al., 2024a*; *Montero-Crespo et al., 2020*). This finding supports the idea that this synaptic characteristic is highly conserved in the human cerebral cortex.

Additionally, the SAS area data of both AS and SS from MEC layers follow a log-normal distribution, which is also manifested in a large number of many functional and morphological properties of the synapses and the circuits that formed, such as postsynaptic current amplitude, dendritic spine volume, and axonal width—or synaptic strength and spike transmission probability (see *Barbour et al., 2007*; *Buzsáki and Mizuseki, 2014*; *Scheler, 2017*). This particular distribution is also consistent with the findings of previous studies performed on the human brain (e.g., *Cano-Astorga et al., 2021*; *Cano-Astorga et al., 2024a*; *Domínguez-Álvaro et al., 2021a*; *Montero-Crespo et al., 2020*). Thus, the fitting of the SAS area to a log-normal distribution likely emerges as a general synaptic characteristic of mammalian cortical synapses.

The analysis of AS SAS area revealed it to be rather uniform across layers, although some significant differences were found in layers I and VI, which had the lowest values (105,906 mm$^2$ and 99,489 mm$^2$, respectively) and layers III and Vab, with the largest AS (130,268 nm$^2$ and 136,111 nm$^2$, respectively). Thus, these data suggest that synaptic size might have a layer-specific component in the MEC. This is of great interest since synaptic size is one the most important parameters correlating with the actual function of the synapse. It has been related to release probability, synaptic strength, efficacy, and plasticity (for review, see *Ganeshina et al., 2004a*; *Ganeshina et al., 2004b*; *Holderith et al., 2012*; *Kharazia and Weinberg, 1999*; *Südhof, 2012*). Moreover, larger synaptic sizes are thought to contain a larger AMPA receptor population, which in turn directly relates to the unitary excitatory postsynaptic potential, thus critically influencing synaptic function (*Araya, 2014*; *Ganeshina et al., 2004a*; *Kharazia and Weinberg, 1999*). In fact, *Palomero-Gallagher and Zilles, 2019*, although using human autopsies samples with postmortem delays between 12 and 18 hr, described a higher density of AMPA receptors in the superficial layers of the EC, which would coincide with the larger SAS area found in layer III. However, it has been reported that some receptors can be found both synaptically and extrasynaptically (*Palomero-Gallagher and Zilles, 2019*). Therefore, caution should be exercised when considering potential correlations between receptor densities, number of synapses, and/or synaptic size.

Furthermore, we observed that the majority of synapses in the MEC, regardless of the analyzed layer, exhibited a macular shape (84%, considering all layers). These findings are consistent with previous research conducted in various brain regions (e.g., *Cano-Astorga et al., 2021*; *Cano-Astorga et al., 2024a*; *Domínguez-Álvaro et al., 2021b*; *Montero-Crespo et al., 2020*). However, differences in this proportion were observed among the MEC layers. Specifically, layer VI had the lowest proportion of macular synapses (78%), for both AS and SS, while layer II-is displayed the highest percentage of macular synapses (88%). Once again, this suggests layer-specific variations. In all layers of the MEC, the sizes of complex-shaped synapses—including perforated, horseshoe-shaped, and fragmented synapses—were consistently larger compared to simple-shaped synapses (macular synapses). As mentioned above, larger synapses contain a higher proportion of AMPA receptors. Complex-shaped synapses have been reported to have a larger population of AMPA and NMDA receptors compared to macular synapses. This makes them a relatively powerful and more stable pool of synapses, contributing to long-lasting memory-related functionality (see *Ganeshina et al., 2004a*; *Ganeshina et al., 2004b*). The smaller area of macular synapses, on the other hand, may play a crucial role in synaptic plasticity (*Kharazia and Weinberg, 1999*). Consequently, the higher proportion of complex-shaped synapses in layer VI may indicate a specific connectivity pattern characterized by synapses with higher synaptic strengths, while the prevalence of macular synapses in layer II, which had the highest macular population, may suggest lower release probability, synaptic strength and efficacy.

## Synapses are mainly established on dendritic spines

Examining the postsynaptic target for each 3D reconstructed synapse may allow us to identify variations in the proportion of synapses on dendritic spines and shafts. These differences might signify a microanatomical specialization of the examined cortical layers. Such variations could have significant functional implications for information processing. Studies have shown (e.g., *Cornejo et al., 2022*) that the membrane potential of the postsynaptic neuron is modulated differently depending on whether a synapse is established on a dendritic spine or a dendritic shaft. Consequently, variations in the proportions of these synaptic targets are likely to have clear functional significance. The present study showed that 60% of total synapses were AS established on dendritic spines (considering all layers), whereas the proportion of AS on dendritic shafts was 34%. Using the same techniques and

analysis (in previous studies), this ratio has varied depending on the region examined. For example, in layers II and III of the EC, the results were similar (54% and 52% of synapses were on dendritic spines, respectively; *Domínguez-Álvaro et al., 2021b*)—and this was also the case in the stratum lacunosum-moleculare of CA1 (57%; *Montero-Crespo et al., 2020*). However, there is a higher proportion of AS on dendritic spines in layer III in Brodmann areas 24, 38 (ventral and dorsal), 21, 17, and 3b (70%, 74%, 72%, 71%, 67%, and 69%, respectively; *Cano-Astorga et al., 2024a*) and in the stratum pyramidale of CA1 (88% in the superficial part and 81% in the deep layer; *Montero-Crespo et al., 2020*). These differences may indicate another microanatomical specialization of the synaptic structure in the neuropil (see *Supplementary file 1o* for a comparison on the distribution of synapses established on the different postsynaptic targets from a variety of mammal species and brain regions).

Moreover, several studies have described a preference of glutamatergic axons (forming AS) for spines and of GABAergic axons (forming SS) for dendritic shafts, in numerous cortical regions and species (reviewed in *DeFelipe et al., 2002*). This finding was also observed in the present study as the proportion of AS:SS established on dendritic spines was 99:1, whereas the proportion of AS:SS on dendritic shafts was 87:13. Considering that the general distribution for AS:SS was 94:6, this shows a clear preference for AS to be established on dendritic spines and SS to be formed on dendritic shafts. This has been consistently observed in every human cortical region and layer analyzed to date using the same techniques and analyses (e.g., *Cano-Astorga et al., 2021*; *Cano-Astorga et al., 2023*; *Cano-Astorga et al., 2024a*; *Domínguez-Álvaro et al., 2021b*; *Montero-Crespo et al., 2020*).

Regarding the size of the synaptic junctions, it has been observed in several neocortical regions that AS on dendritic spine heads are larger than those on dendritic shafts or dendritic spine necks (*Cano-Astorga et al., 2023*; *Cano-Astorga et al., 2024a*). However, in the CA1 field of the hippocampus, the AS on shafts were slightly larger than axospinous AS (which include both spine heads and necks), although the differences were not found to be statistically significant (*Montero-Crespo et al., 2020*). In the case of the MEC, however, we observed that AS, whether established on dendritic spine heads or dendritic shafts, shared a similar size, whereas AS on dendritic spine necks were significantly smaller. This suggests another specialization of synaptic anatomical features that may vary between cortical areas.

Additionally, the study of the multisynaptic spines revealed that, regardless of the layer analyzed, the vast majority of spines contained only a single AS on their head (96%). Once again, this percentage is in agreement with previous studies that showed around 95% of AS on spines were single synapses in layer III of Brodmann areas 21, 24, and 38 (ventral and dorsal), 17, 4 and 3b (*Cano-Astorga et al., 2024a*). The hippocampal CA1 region presented a slightly higher percentage of single AS on spines (97.9%; *Montero-Crespo et al., 2020*). These results highlight another microanatomical specialization concerning the presence of single AS on spines, although its functional relevance remains unclear. In fact, the generation of multisynaptic spines has been associated with synaptic potentiation and has been linked to memory processes (*Giese et al., 2015*; *McLeod et al., 2020*). Furthermore, in the mouse neocortex, it has been suggested that spines receiving one AS and one SS are electrically more stable than spines establishing a single AS (*Villa et al., 2016*). As mentioned above, more stable synapses would lead to an increment in their synaptic size. Indeed, in the MEC, AS established on multisynaptic spines containing an SS were 30% larger than single AS.

## Complex-shaped synapses are more frequently found on dendritic spines

In the present study, we observed that, out of the total macular AS, the distribution of these synapses on dendritic spine heads and dendritic shafts was 61:39. This ratio is very similar to the distribution of all AS (without considering their shape) on the same postsynaptic targets, which was 63:37. However, when making the same comparison for complex synapses, this ratio was 75:25 for AS established on dendritic spine heads and dendritic shafts. This indicates that the proportion of complex synapses in spine heads is almost 15% higher than what would be expected based on the overall distribution of synapses.

Indeed, complex AS established on dendritic spine heads were larger than those established on dendritic shafts, whereas macular AS established on dendritic shafts were larger than those on dendritic spine heads (see *Figure 7*). As mentioned above, complex-shaped synapses presented larger synaptic sizes than macular synapses, and this correlates with higher synaptic strengths. This

type of synapse contributes to the formation of more stable circuits (*Ganeshina et al., 2004a*; *Ganeshina et al., 2004b*). In the MEC, while the general distribution of synapses established on dendritic spine heads and dendritic shafts appears to be quite similar, especially when compared to other neocortical regions such as Brodmann areas 21, 24, 38, 17, 4 or 3b (see *Cano-Astorga et al., 2024a*), complex synapses established on dendritic spine heads may play a crucial role in consolidating the synaptic circuits that are formed in the different layers. Indeed, persistent dendritic spines, which form the basis of consolidated circuits, are described to be larger and have bigger PSDs (see *Holtmaat et al., 2006*; *Yuste, 2011*).

## Ultrastructural characteristics of synapses are similar across layers of the MEC

Similarity of the synaptic characteristics of the MEC may seem surprising given its cytoarchitectonic and innervation complexity. Indeed, MEC layers contain a variety of neuronal classes heterogeneously distributed across layers, as extensively reviewed in *Kobro-Flatmoen and Witter, 2019*. All these neurons are highly innervated, receiving information from both local and intercortical connections. *Figure 8* summarizes the extrinsic and intrinsic circuits formed in the different layers of the MEC. Unfortunately, all these connectivity data come from experimental animals, and information from the human brain is very scarce in this regard. Nevertheless, *Figure 8* illustrates clear differences in the connectivity pattern between the different layers, where superficial neurons mainly receive information from other brain areas (such as the PRC and PHC) to be transmitted to the hippocampus, whereas deep-located cells are the recipient of the hippocampal return connections, giving rise to the cortical output of the EC (*Insausti and Amaral, 2008*; *Insausti and Amaral, 2012*; *Melzer et al., 2012*; *Muñoz and Insausti, 2005*; *Nilssen et al., 2019*; *Ohara et al., 2021*; *Ohara et al., 2023*; *Sürmeli et al., 2016*; *Vandrey et al., 2022*). In addition, the MEC shows profuse intracortical connectivity, with (i) deep-located neurons sending projections to upper layers (*Chrobak and Amaral, 2007*; *Ohara et al., 2021*; *Sürmeli et al., 2016*) and (ii) both excitatory and inhibitory intralayer innervation (*Nilssen et al., 2019*; *Rozov et al., 2020*; *Winterer et al., 2017*).

The analysis of MEC layer II is of particular interest. This layer is characterized by the presence of reelin-positive stellate cells and calbindin-positive pyramidal neurons clustered together to form 'cell islands' with gaps between them that contain almost no neurons (*Insausti et al., 2017*; *Kobro-Flatmoen and Witter, 2019*). In the present study, we were able to selectively analyze the neuropil of both zones independently. Our data showed no differences in any parameter studied. In fact, these results were very similar to those described in one of our previous studies performed in the same layer on different autopsy samples, but with no distinction made between islands and non-island areas (*Domínguez-Álvaro et al., 2021a*). For example, in this study, a density of 0.40 synapses/μm$^3$ was found in EC layer II, which is very close to the 0.42 and 0.44 synapses/μm$^3$ found in the present study in layer II, in the island and non-island regions, respectively.

Conversely, layer VI stood out in many of the synaptic parameters analyzed. Although the synaptic density in this layer was comparable to that of other layers, overall it had the smallest synapses and both the smallest macular and complex-shaped AS. It also had the lowest proportion of macular synapses (78%) and the highest proportion of synapses established on dendritic spine heads (76%). In humans, layer VI contains a variety of cell types, ranging from small- to medium-sized bipolar and multipolar calbindin-positive cells, large pyramidal calretinin-positive neurons, and multipolar parvalbumin interneurons, among others (*Kobro-Flatmoen and Witter, 2019* and references wherein). Additionally, this layer receives projections from numerous regions, including the hippocampus, the medial frontal and orbitofrontal cortex, the thalamus, and the claustrum (see *Figure 8*). The synaptic peculiarities described in the present study may have a critical impact on how this layer processes the information it receives.

Despite the complexity of the MEC in terms of its cytoarchitecture, connectivity, and function, it is remarkable that the ultrastructural synaptic characteristics remain rather similar across layers. One possible explanation is that several of the parameters investigated in this study (e.g., synaptic density, proportion of AS:SS) do not depend on the cytoarchitecture or connectivity of the region, at least in the case of the MEC neuropil. Very few papers have analyzed the synaptic characteristics at the ultrastructural level of different layers in the same human brain region. However, *Montero-Crespo et al., 2020* found that the ultrastructural synaptic characteristics of the human hippocampus

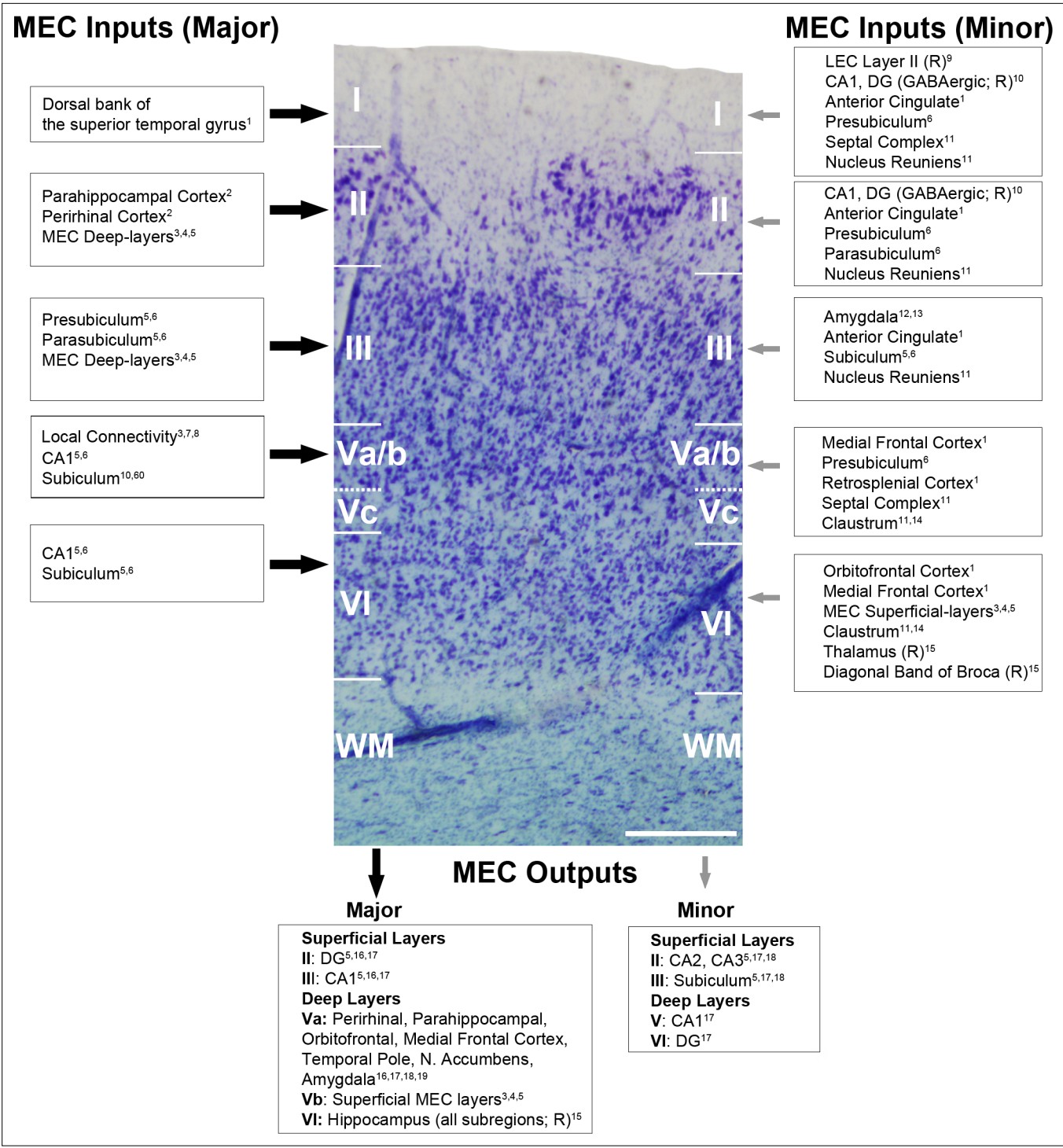

**Figure 8.** Main extrinsic and intrinsic connections of the MEC. Major and minor MEC inputs and outputs have been represented with large (black) and small (gray) arrows, respectively. Connections have been illustrated on a photograph of a Nissl-stained coronal section from the human MEC, where all layers can be individually identified. The data mainly come from primates and rodents (marked as 'R' in the figure, for those connections only described in rodents). Scale bar indicates 415 μm. References: [1]*Insausti and Amaral, 2008*; [2]*Maass et al., 2015*; [3]*Chrobak and Amaral, 2007*; [4]*Sürmeli et al., 2016*; [5]*Nilssen et al., 2019*; [6]*Witter and Amaral, 2021*; [7]*Gerlei et al., 2021*; [8]*Rozov et al., 2020*; [9]*Vandrey et al., 2022*; [10]*Melzer et al., 2012*; [11]*Insausti et al., 1987*; [12]*Pitkänen et al., 2002*; [13]*Roesler and McGaugh, 2022*; [14]*Kitanishi and Matsuo, 2017*; [15]*Ben-Simon et al., 2022*; [16]*Insausti and Amaral, 2012*; [17]*Witter and Amaral, 1991*; [18]*Muñoz and Insausti, 2005*; [19]*Ohara et al., 2021*.

may be influenced by the innervation that the different layers receive. The ultrastructural synaptic characteristics in the layers receiving CA3 Schaffer collaterals (e.g., stratum pyramidale and stratum radiatum) differ from those in stratum lacunosum moleculare, which mainly receives information from the EC. Similarly, *Cano-Astorga et al., 2023*—when analyzing the ultrastructural synaptic characteristics of layer III in different cingulate and temporopolar cortices—described remarkable differences in synaptic size and density between the ventral and dorsal part of BA38, which may also be correlated with the differential connectivity pattern exhibited by this brain area.

Finally, another crucial aspect is the interindividual variability of human samples. There are numerous studies describing interindividual variability in the human cerebral cortex, both in terms of structure and function (e.g., *DeFelipe, 2015*). However, in the present study, the synaptic parameters analyzed showed relatively little variability (see *Supplementary file 1b,d,g,i,k, and m*, which provide data from all MEC layers for each individual case). Indeed, the proportion of AS:SS showed remarkable uniformity across individuals in all layers (*Table 1*). This apparent similarity was further confirmed through the examination of the variability between cases (*Table 2*, *Figure 1—figure supplement 2*, *Figure 3—figure supplement 1*, *Figure 4—figure supplements 4 and 5*; *Supplementary file 3*) and stacks of images (*Supplementary file 1n*), supporting the regularity of the data in all layers of the MEC. Furthermore, in all individuals, layers, and stack of images, SS had significantly smaller sizes and were in much lower numbers than AS. This is in line with the fact that, as already discussed, synaptic characteristics of SS are similar in the different cerebral regions, layers, and animals examined so far (e.g., *Cano-Astorga et al., 2023*; *Cano-Astorga et al., 2024a*).

Overall, the present data point to a set of highly conserved synaptic characteristics in all layers of the MEC, while some data point to a laminar specificity. Future studies on human synaptic nanoconnectivity of different layers in additional brain regions could confirm whether there is a commonly shared trend across the human cortex.

## Materials and methods
### Tissue preparation
Human brain tissue was obtained from autopsies from two men and one woman (with short post-mortem delays of less than 4 hr, and ages ranging from 40 to 66 years old; *Supplementary file 1p*) with no recorded neurological or psychiatric alterations (supplied by Unidad Asociada Neuromax, Laboratorio de Neuroanatomía Humana, Facultad de Medicina, Universidad de Castilla-La Mancha, Albacete, Spain). The consent of the individuals was obtained, and the sampling procedure was approved by the Institutional Ethical Committee of the Albacete University Hospital. The tissue was obtained following national laws and international ethical and technical guidelines on the use of human samples for biomedical research purposes. Human brain tissue from the same subjects has been used in previous studies (e.g., *Cano-Astorga et al., 2021*; *Cano-Astorga et al., 2023*; *Domínguez-Álvaro et al., 2021a*; *Domínguez-Álvaro et al., 2021b*; *Montero-Crespo et al., 2020*).

Upon removal, brains were immediately immersed in cold 4% paraformaldehyde (Sigma-Aldrich, St Louis, MO) in 0.1 M phosphate buffer (PB; Panreac, 131965, Spain), pH 7.4 for 24–48 hr and sectioned into 1.5-cm-thick coronal slices. Small blocks of the cortex (15 × 10 × 10 mm) were then transferred to a second solution of 4% paraformaldehyde in PB for 24 hr at 4°C.

After fixation, the tissue was washed in PB and sectioned coronally in a vibratome (Vibratome Sectioning System, VT1200S Vibratome, Leica Biosystems, Germany). Sections containing all layers of the EC were selected and processed for Nissl staining to determine the cytoarchitecture (*Figure 9*). In addition, immunostaining for anti-PHF-Tau and anti-Aβ (performed in adjacent sections to those processed for EM) revealed neither the presence of Aβ-plaques nor PHF-Tau neurons in the analyzed individuals.

### Electron microscopy
Selected sections (150-µm-thick) were post-fixed for 24 hr in a solution containing 2% paraformaldehyde, 2.5% glutaraldehyde (TAAB, G002, UK), and 0.003% $CaCl_2$ (Sigma, C-2661-500G, Germany) in sodium cacodylate (Sigma, C0250-500G, Germany) buffer (0.1 M). The sections were treated with 1% $OsO_4$ (Sigma, O5500, Germany), 0.1% potassium ferrocyanide (Probus, 23345, Spain), and 0.003% $CaCl_2$ in sodium cacodylate buffer (0.1 M) for 1 hr at room temperature. They were then stained with

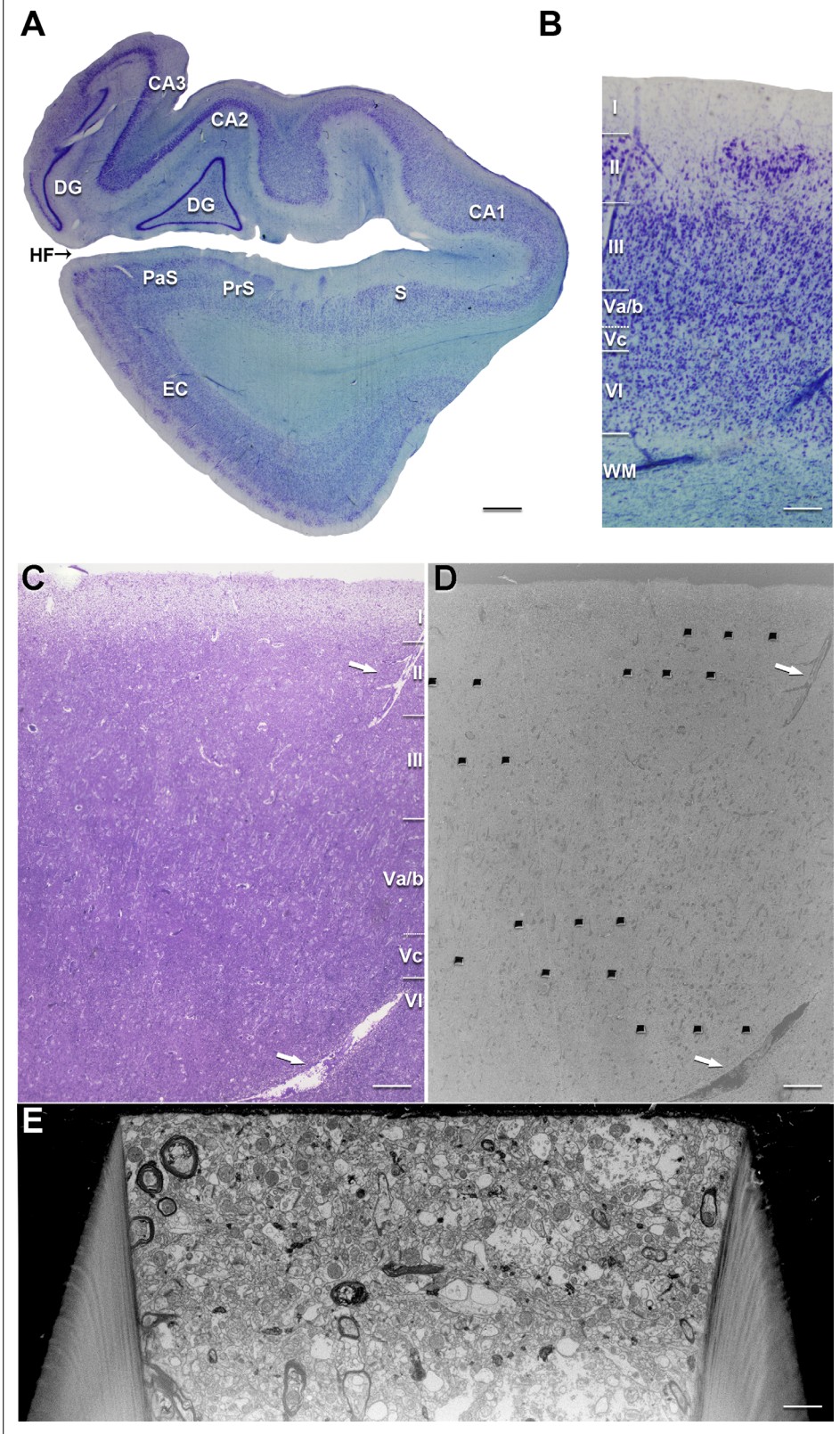

**Figure 9.** Correlative light/electron microscopy of the MEC layers. (**A**) Low-power photograph of a coronal section from the human hippocampal formation and entorhinal cortex. (**B**) Higher magnification of the MEC shown in (**A**), to illustrate the laminar pattern (layers I to VI are indicated). This image was reused from *Figure 8*. The delimitation of layers is based on the toluidine blue-stained semithin section (**C**), adjacent to the block for focused ion beam

*Figure 9 continued*

FIB/SEM imaging (**D**). (**D**) SEM image illustrating the block surface with trenches made in the neuropil of MEC layers. Arrows in (**A**) and (**B**) mark the same blood vessel, allowing the regions of interest to be accurately located. (**E**) SEM image showing the front of a trench made to acquire the FIB/SEM stack of images. CA1: cornu ammonis 1; CA2: cornu ammonis 2; CA3: cornu ammonis 3; DG: dentate gyrus; EC: entorhinal cortex; HF: hippocampal fissure; PaS: parasubiculum; PrS: presubiculum; S: subiculum. Scale bar: 1.4 mm in (**A**), 260 µm in (**B**), 175 µm in (**C**), 165 µm in (**D**) and 2 µm in (**E**).

1% uranyl acetate (EMS, 8473, USA), dehydrated, and flat-embedded in Araldite (TAAB, E021, UK) for 48 hr at 60°C (*Merchán-Pérez et al., 2009*; *Cano-Astorga et al., 2024b*). The embedded sections were then glued onto a blank Araldite block and trimmed. Semithin sections (1–2 µm thick) were obtained from the surface of the block and stained with 1% toluidine blue (Merck, 115930, Germany) in 1% sodium borate (Panreac, 141644, Spain). The last semithin section (which corresponds to the section immediately adjacent to the block surface) was examined under a light microscope and photographed to accurately locate the neuropil region to be later examined with the FIB/SEM (*Figure 9*).

## Volume fraction estimation of cortical elements

Five to six semithin sections (1–2 µm thick) from each case, stained with 1% toluidine blue, were used to estimate the volume fraction occupied by neuropil, cell bodies (including neuronal somata, glia, and undetermined cells), and blood vessels. This estimation was performed applying the Cavalieri principle (*Gundersen et al., 1988*) by point counting (Q) using the integrated Stereo Investigator stereological package (version 8.0, MicroBrightField Inc, VT) attached to an Olympus light microscope (Olympus, Bellerup, Denmark) at ×40 magnification. A grid, whose points covered an area of 2500 µm$^2$, was randomly overlaid at 10–15 sites over the previously traced layer (separately) in each semithin section to determine the volume fraction (Vv) occupied by the different elements. Vv (e.g., in the case of the neuropil) was estimated with the following formula: Vv-neuropil = Q-neuropil × 100/ (Q-neuropil + Q-neurons + Q-glia + Q-undetermined cells + Q-blood vessels).

## Three-dimensional electron microscopy

The blocks containing the embedded tissue were glued onto a sample stub using carbon adhesive tabs (Electron Microscopy Sciences, 77825-09, USA). All surfaces of the blocks except the one to be studied (the top surface) were covered with silver paint (Electron Microscopy Sciences, 12630) to prevent any charging of the resin. The stubs with the mounted blocks were then placed into a sputter coater (Emitech K575X, Quorum Emitech, Ashford, Kent, UK) and the top surface was coated with several 10-nm-thick layers of gold/palladium to facilitate charge dissipation.

The 3D study of the samples was carried out using a dual beam microscope (Crossbeam 40 electron microscope, Carl Zeiss NTS GmbH, Oberkochen, Germany). This instrument combines a high-resolution field-emission SEM column with a focused gallium ion beam (FIB), which permits the removal of thin layers of material from the sample surface on a nanometer scale. Using a 7 nA ion current (30 kV accelerating voltage), a first coarse trench was milled to have visual access to the tissue below the block surface (*Figure 9*). The exposed surface of this trench was then milled with a 100 pA ion current (30 kV accelerating voltage) to remove 20 nm. As soon as one layer of material is removed by the FIB, the exposed surface of the sample is imaged by the SEM using the backscattered electron detector (at 1.7 kV acceleration potential). The sequential automated use of FIB milling and SEM imaging allowed us to obtain long series of photographs of a 3D sample of selected regions (*Merchán-Pérez et al., 2009*). Image resolution in the xy plane was 5 nm/pixel. Resolution in the z-axis (section thickness) was 20 nm, and the image size was 2048 × 1536 pixels. These parameters ensured an optimal field of view where the different types of synaptic junctions could still be clearly identified in a reasonable period of time (approximately 12 hr per stack of images). Although the sample is destroyed during milling, the size of the trench needed to acquire a stack of serial images is around 30–40 microns wide, allowing the collection of multiple stacks from the same region.

Alignment (registration) of serial microphotographs obtained with the FIB/SEM was performed with the 'Register Virtual Stack Slices' plug-in in FIJI, which is a version of ImageJ with a collection of preinstalled plug-ins (https://fiji.sc/; *Schindelin, 2012*). To avoid deformation of the original

images, as well as size changes and other artifacts, we selected a registration technique that only allows translation of individual images, with no rotation.

All measurements were corrected for tissue shrinkage, which occurs during the tissue and electron microscopy processing (*Merchán-Pérez et al., 2009*). To estimate the shrinkage in our samples, we photographed and measured the area of the vibratome sections with ImageJ (ImageJ 1.51; NIH, USA), both before and after processing for electron microscopy. The section area values after processing were divided by the values before processing to obtain the volume, area, and linear shrinkage factors, yielding correction factors of 0.90, 0.93, and 0.97, respectively. Nevertheless, in order to compare with previous studies—in which no correction factors had been included or such factors were estimated using other methods—in the present study, we provide both sets of data (i.e., corrected and not corrected for shrinkage).

Additionally, a correction in the volume of the stack of images for the presence of fixation artifacts (e.g., swollen neuronal or glial processes) was applied after quantification with the Cavalieri principle (*Gundersen et al., 1988*). A total of 63 FIB/SEM stacks were examined using FIJI (https://fiji.sc/), and the artifact volume ranged from 1.4 to 27.0% of the volume stacks.

## Three-dimensional analysis of synapses

The 63 stacks of images obtained by the FIB/SEM were analyzed using EspINA software (EspINA Interactive Neuron Analyzer, 2.9.12; https://cajalbbp.csic.es/espina-2), allowing the segmentation of synapses in the reconstructed 3D volume (see *Morales et al., 2011* for a detailed description of the segmentation algorithm). Of these stacks, six were from layers II and III of case M16, which were previously used in *Domínguez-Álvaro et al., 2021a* and were reanalyzed in the present study.

The user identifies synapses by assessing the presence of pre- and postsynaptic densities, along with the accumulation of synaptic vesicles in the presynaptic terminal. Synaptic segmentation depends on the intense electron density of these pre- and postsynaptic regions. Users must establish a gray-level threshold, and the segmentation algorithm subsequently selects pixels in the synaptic junction darker than the chosen threshold. The resulting segmentation creates a 3D object encompassing the active zone and postsynaptic densities as these represent the two darkest areas of the synaptic junction. EspINA software allows the user to supervise the result of segmentation and reconstruction; each synapse segmentation is revised and validated by the user, and the segmentation can be manually edited if necessary to accurately delimit the active zone and PSD of each synapse.

There is a consensus for classifying cortical synapses into AS (or type I) and SS (or type II). The main characteristic distinguishing synapses is the prominent or thin PSD, respectively (*Colonnier, 1968*; *Gray, 1959*; *Peters et al., 1991*; *Figure 10*).

Also, these two types of synapses are associated with different functions: AS are mostly glutamatergic and excitatory, while SS are mostly GABAergic and inhibitory (*DeFelipe and Fariñas, 1992*; *Ascoli et al., 2008*). Nevertheless, in single sections, the synaptic cleft and the pre- and postsynaptic densities are often blurred if the plane of the section does not pass at right angles to the synaptic junction. Since the software EspINA allows navigation through the stack of images, it was possible to unambiguously identify every synapse as AS or SS based on the thickness of the PSD (*Merchán-Pérez et al., 2009*; *Cano-Astorga et al., 2024b*). Synapses with prominent PSDs were classified as AS, while thin PSDs were classified as SS (*Figure 10*). All synapses were manually identified by an expert, and unclear synapses were reevaluated by the consensus of two or three experts.

EspINA provided the 3D reconstruction of every synaptic junction and permitted the application of an unbiased 3D CF, formed by a rectangular prism enclosed by three acceptance planes and three exclusion planes marking its boundaries. All synapses within the CF were counted, as were those intersecting any of the acceptance planes (*Figure 11*). Synapses that were outside the CF, or intersecting any of the exclusion planes, were not counted. Hence, the number of synapses per unit volume was calculated directly by dividing the total number of synapses counted by the volume of the CF (*Merchán-Pérez et al., 2009*).

Synaptic size was calculated using the SAS, which was automatically extracted by EspINA (*Figures 11 and 12*). In the present study, we have used the area of the SAS (which comprises both the active zone and the PSD, *Morales et al., 2013*) as a measurement of the synaptic size.

In addition, the visualization of each 3D reconstructed synaptic junction allowed us to determine the synaptic morphology, based on the presence of perforations or indentations in the junction

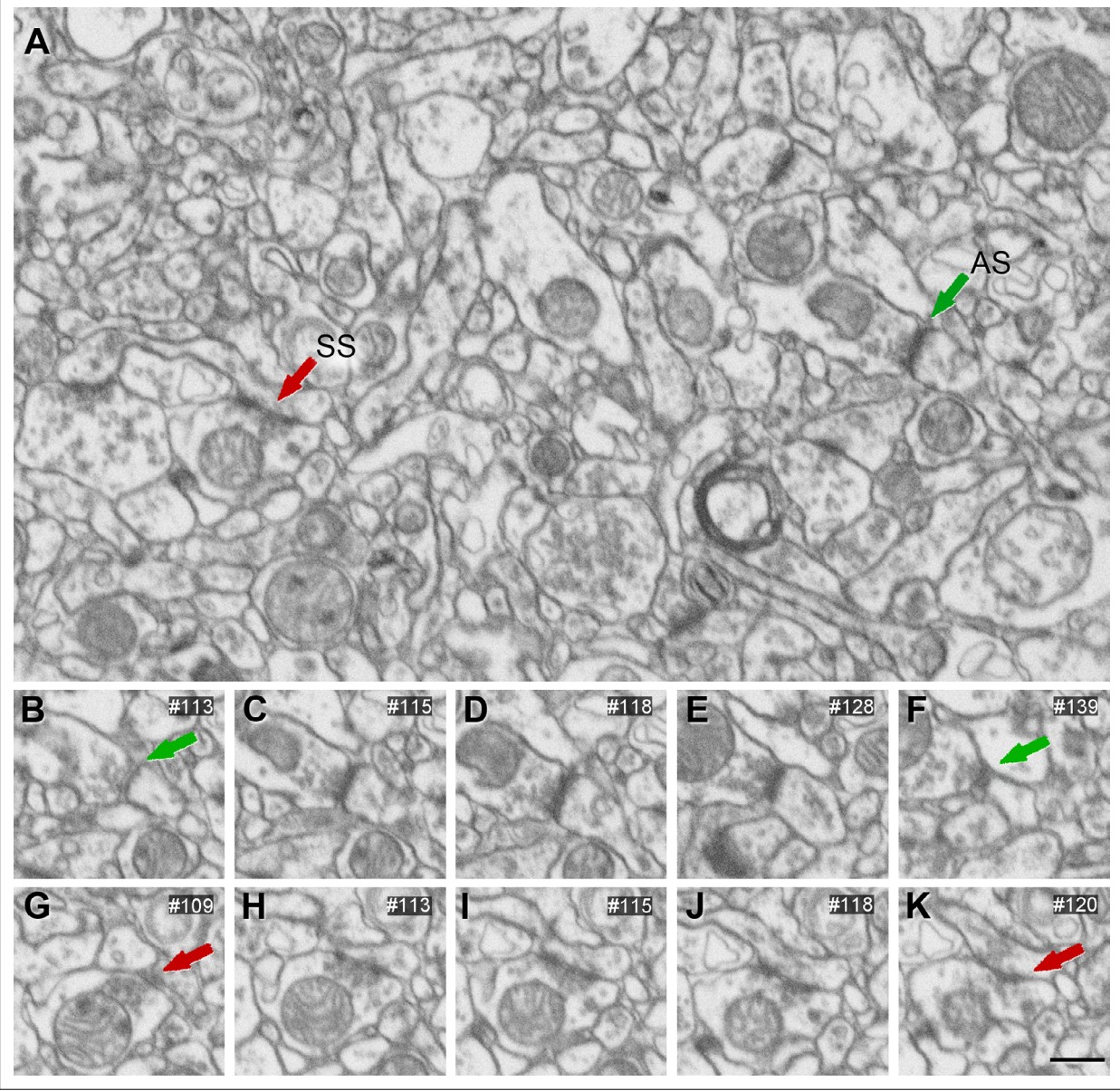

**Figure 10.** Focused ion beam FIB/SEM images from layer Va/b. (**A**) Two synapses are indicated as examples of asymmetric (AS, green arrow) and symmetric (SS, red arrow) synapses. *Figure 10—figure supplements 1–6* show FIB/SEM images from the neuropil of the rest of the layers. (**B–F**) FIB/SEM serial images of the AS indicated in (**A**). (**G–K**) FIB/SEM serial images of the SS indicated in (**A**). Synapse classification was based on the examination of full sequences of serial images. Scale bar (in **K**): 500 nm for (**A–K**).

The online version of this article includes the following figure supplement(s) for figure 10:

**Figure supplement 1.** Ultrastructure of the neuropil in layer I.

**Figure supplement 2.** Ultrastructure of the neuropil in layer II-is.

**Figure supplement 3.** Ultrastructure of the neuropil in layer II-ni.

**Figure supplement 4.** Ultrastructure of the neuropil in layer III.

**Figure supplement 5.** Ultrastructure of the neuropil in layer Vc.

**Figure supplement 6.** Ultrastructure of the neuropil in layer VI.

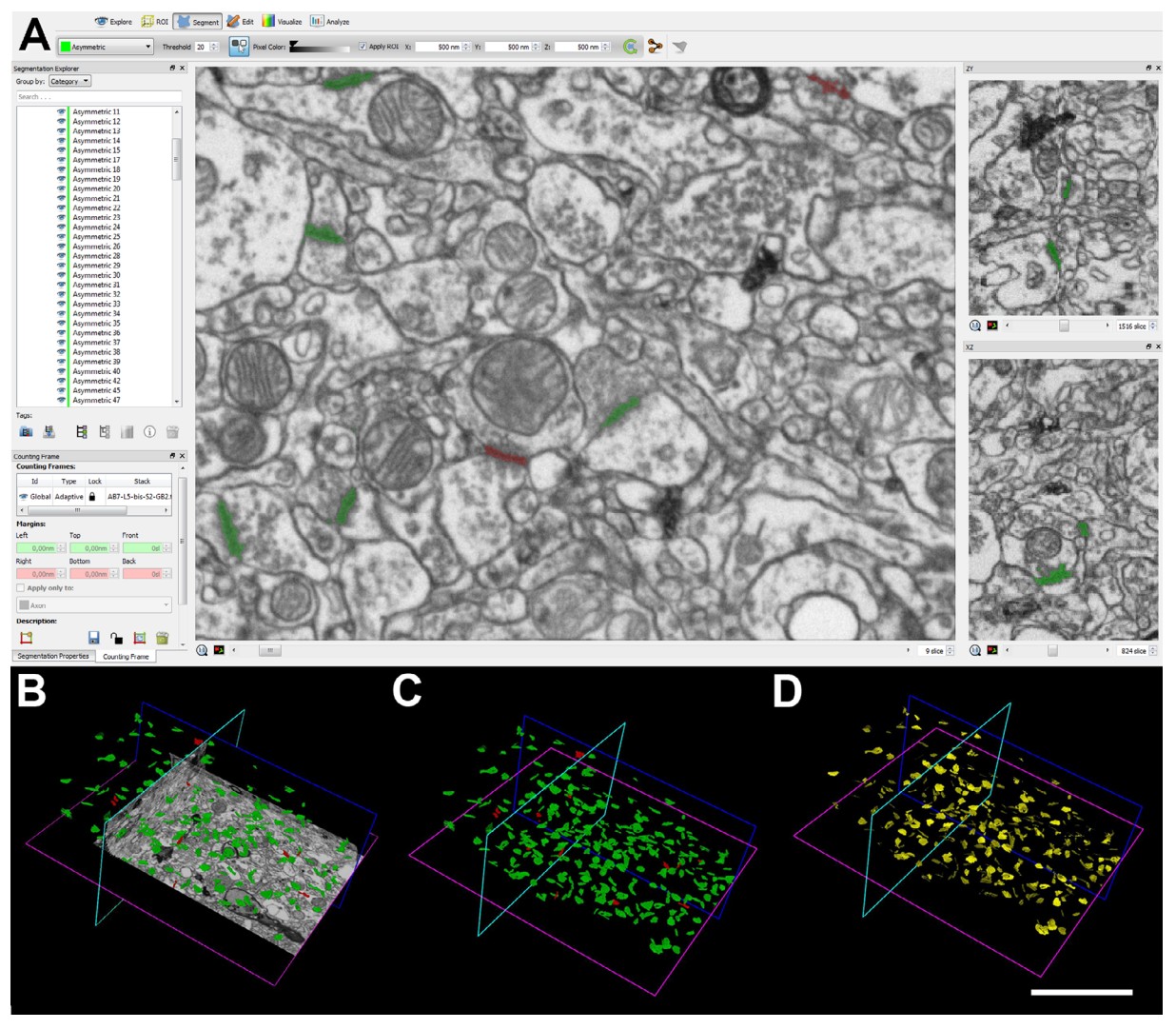

**Figure 11.** Screenshots of the EspINA software user interface. (**A**) In the main window, sections are viewed through the xy plane, and the other two orthogonal planes, yz and xz, are shown in adjacent windows (right). (**B–D**) 3D views showing: the three orthogonal planes and the 3D reconstruction of asymmetric synapses (AS) (green) and symmetric synapses (SS) (red) segmented synapses (**B**); only the reconstructed synapses (**C**); and the synaptic apposition surface (SAS) for each reconstructed synapse (**D**, in yellow). Scale bar (in **D**): 11 µm for (**B–D**).

perimeters. In this regard, the synapses could be classified into four types: macular (with a flat, disk-shaped PSD), perforated (with one or more holes in the PSD), horseshoe (with an indentation in the perimeter of the PSD) or fragmented (with two or more physically discontinuous PSDs). This classi-fication has been used consistently in other studies, yielding a comprehensive comparison between human brain areas (e.g., *Cano-Astorga et al., 2023*).

Furthermore, to identify the postsynaptic targets of the axon terminals, we navigated through the image stacks using EspINA to determine whether the postsynaptic element was a dendritic spine or a dendritic shaft (*Figure 13* and *Figure 13—video 1*). As previously described in *Cano-Astorga et al., 2023*, unambiguous identification of dendritic spines requires the spine to be visually followed to the parent dendrite, in which case we refer to it as a 'complete spine'. When synapses are established on a dendritic spine-shaped postsynaptic element whose neck cannot be traced to the parent dendrite, we identify these elements as 'incomplete spines'. These incomplete dendritic spines were identi-fied based on their size and shape, the lack of mitochondria, and the presence of a spine apparatus (*Peters et al., 1991*)—or because they were filled with a characteristic fluffy material (used to describe the fine and indistinct filaments present in the spines). We also described the presence of single or multiple synapses on a single dendritic spine. Finally, if the synapse was established on a dendritic

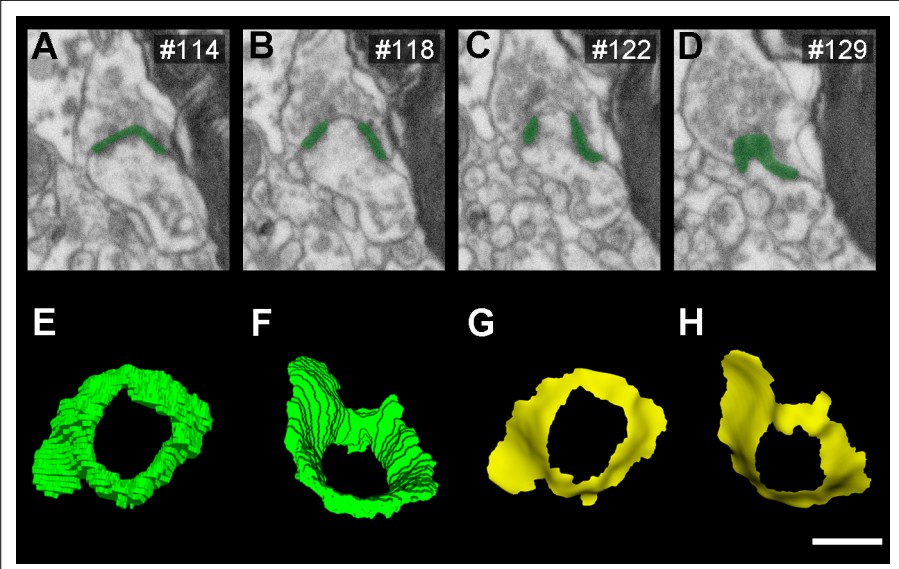

**Figure 12.** Extraction of the synaptic apposition surface (SAS). (**A–D**) An example of focused ion beam FIB/SEM serial images showing an asymmetric synapses (AS) (in green). (**E, F**) 3D Reconstruction of the AS, indicated in (**A–D**), in different rotation planes. A central perforation can be observed. (**G, H**) Automatically generated SAS of the reconstructed synapse in the same rotation planes as in (**E, F**). The obtained SAS reproduces the profile and the irregular curvature of the 3D reconstructed synapse, in which the same central perforation can be observed. SAS includes both the active zone and the postsynaptic density (PSD) of the 3D segmented synapse. Scale bar (in **H**): 500 nm in (**A–D**); 165 nm in (**E–H**).

shaft, we could further determine whether the target dendrite was spiny or aspiny (based on the presence of dendritic spines along its path throughout the image stack).

## Spatial distribution analysis of synapses

Spatial distribution features (centroids) of each 3D reconstructed synapse were also calculated by EspINA in all FIB/SEM stacks of images. To analyze the spatial distribution of synapses (i.e., whether synapses are arranged in a regular, random, or clustered distribution), spatial point-pattern analysis was performed as described elsewhere (*Merchán-Pérez et al., 2014*). Briefly, we compared the actual position of the synapse centroids with the CSR model—a random spatial distribution model that defines a situation where a point is equally likely to occur at any location within a given volume. The mathematical model underlying CSR is a homogeneous spatial Poisson point process. For each of the 63 different samples, we calculated three functions commonly used for spatial point pattern analysis: F, G, and K functions, each of which computes different parameters (see *Merchán-Pérez et al., 2014* for a detailed description). Briefly, the F function, known as the empty space function, is—for a distance d—the probability that the distance of each point to its nearest synapse centroid is at most d. The G function, also called the nearest-neighbor distance cumulative distribution function, is—for a distance d—the probability that a typical point separates from its nearest neighbor by a distance of at most d. The K function, called the reduced second-moment function, is—for a distance d—the expected number of points within a distance d of a typical point of the process divided by the intensity $\lambda$. An estimation of the K function is given by the mean number of points within a sphere of increasing radius d centered on each sample point, divided by an estimation of the expected number of points per unit volume. In addition, we calculated the theoretical distribution that corresponds to a homogeneous spatial Poisson point process which serves as a reference point, in each stack of images. We then calculated the envelope area corresponding to the Poisson distribution, obtained from 99 simulations performed with the same number of centroids in each of the stacks analyzed. Thus, if the curves obtained from the F, G, or K functions in any or the stacks of images appear to be inside the envelope area, it means that the spatial distribution of the centroids in the stack of images is indistinguishable from random. This study was carried out using the spatstat package and R Project program (*Baddeley et al., 2016*).

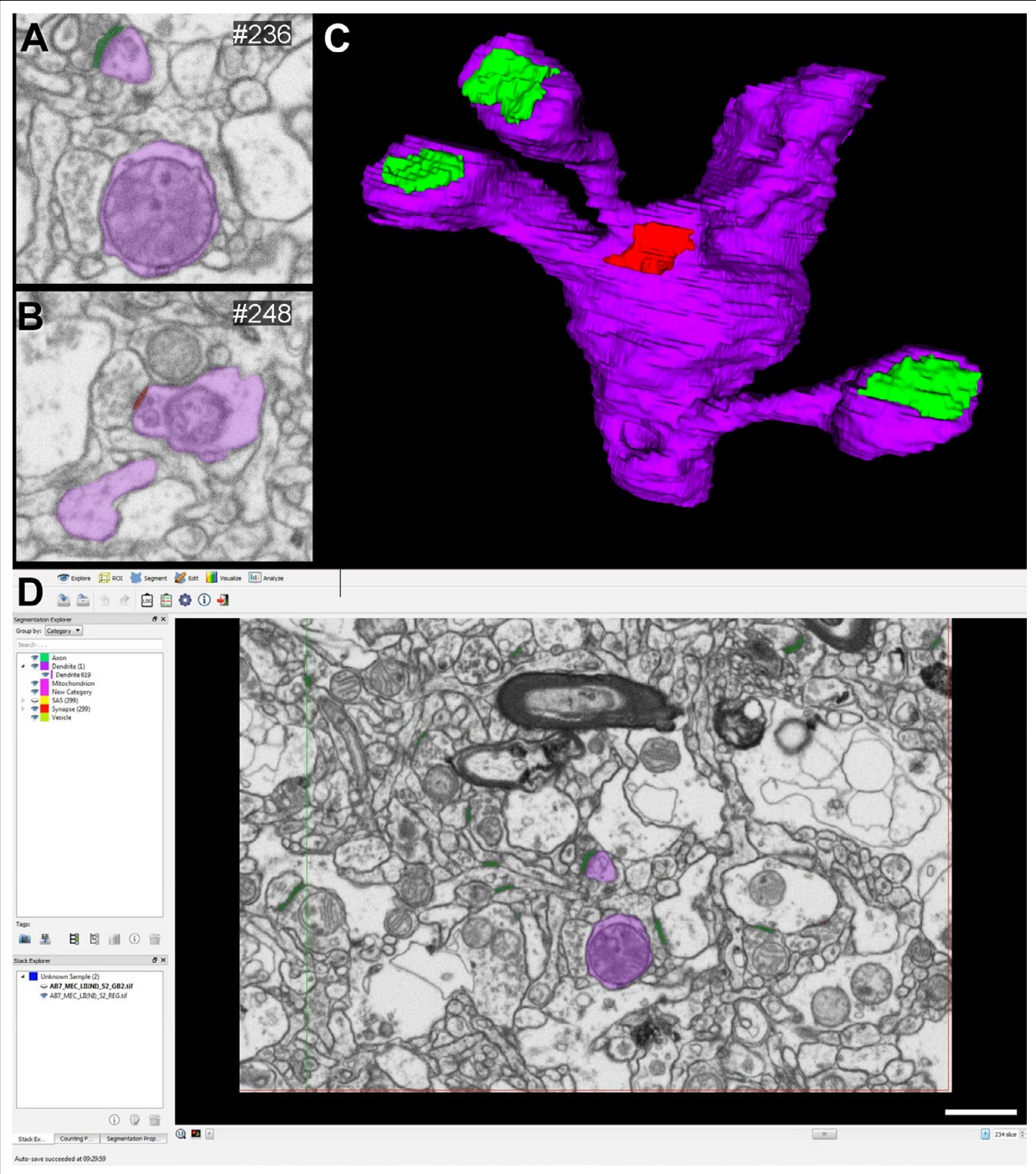

**Figure 13.** 3D reconstruction of a dendritic segment from focused ion beam FIB/SEM serial images. (**A, B**) Serial images showing a dendritic segment partially reconstructed (in purple). An asymmetric synapse (in **A**) on a dendritic spine head and a symmetric synapse (in **B**) on a shaft are indicated in green and red, respectively. (**C**) 3D reconstruction of the dendritic segment indicated in (**A**) and (**B**). Three dendritic spines receiving asymmetric synapses can be observed, along with a symmetric synapse formed on the dendritic shaft. (**D**) Snapshot of EspINA software interface displaying the reconstructed dendritic segment. Scale bar (in **C**): 500 nm in (**A**, **B**), 370 nm in (**C**) and 900 in (**D**).

The online version of this article includes the following video for figure 13:

**Figure 13—video 1.** Visualization of the reconstructed dendritic segment of *Figure 13* using EspINA software.
https://elifesciences.org/articles/96144/figures#fig13video1

## Statistical analysis

We studied differences in synaptic density, as well as morphological and spatial parameters, between layers by conducting multiple mean comparison tests on data from the various MEC layers studied. In this case, we performed KW, followed by a post hoc Dunn's multiple comparison test (when KW test finds statistical significance), and MW nonparametric $U$-test, since normality and homoscedasticity criteria were not met in any comparison. Frequency distribution analysis of the area of the SAS was performed using KS nonparametric test. The analysis of proportions was carried out using pairwise comparisons in contingency tables ($\chi^2$), followed by a post hoc Bonferroni correction test for multiple comparisons. The criterion for statistical significance was considered to be met if $p<0.05$ when the sample size was equal to the number of subjects or image stacks (i.e., MW and KW tests) and if $p<0.0001$ when the sample size was equal to the number of synapses (i.e., KS and pairwise comparison) in order to avoid overestimation of the differences due to a very large sample size.

Data variability between individuals and stacks of images was estimated by calculating the CV of each synaptic parameter in every layer. CV was calculated by dividing the SD by the mean for each of the synaptic parameters analyzed (*Table 2*; *Supplementary file 1n*). Additional statistical comparisons of the variability between individuals were carried out using KW, followed by a post hoc Dunn's multiple comparison test (*Supplementary file 3*).

All statistical studies were performed using GraphPad Prism statistical package (Prism 9 for Windows, GraphPad Software Inc, USA) and R Project software (R.4.2.2; Bell Laboratories, NJ; http://www.R-project.org). Additionally, goodness-of-fit analysis for the different SAS distributions in each layer was performed using Easyfit Professional 5.5 (MathWave Technologies).

## Acknowledgements

We would like to thank Unidad de Tecnologías Ómicas at Instituto Cajal for its useful statistical analysis service; L Valdés and C Álvarez for their technical assistance; and Nick Guthrie for his excellent text editing. Grant PID2021-127924NB-I00 funded by MCIN/AEI/10.13039/501100011033 (JD). CSIC Interdisciplinary Thematic Platform—Cajal Blue Brain (JD). European Union's Horizon 2020 Framework Programme for Research and Innovation Specific Grant Agreement No. 945539 (Human Brain Project SGA3) (JD). Fellowship Grant PRE2019-089228 funded by MCIN/AEI/10.13039/501100011033 (NC-A). Fellowship Grant FPU19/00007 funded by Spanish Ministry of Universities (SP-A)

## Additional information

### Funding

| Funder | Grant reference number | Author |
|---|---|---|
| Ministerio de Ciencia e Innovación | PID2021-127924NB-I00 | Javier DeFelipe |
| CSIC Interdisciplinary Thematic Platform | Cajal Blue Brain | Javier DeFelipe |
| Horizon 2020 Framework Programme | 10.3030/945539 | Javier DeFelipe |
| Ministerio de Ciencia e Innovación | PRE2019-089228 | Nicolas Cano-Astorga |
| Ministerio de Ciencia, Innovación y Universidades | FPU19/00007 | Sergio Plaza-Alonso |

The funders had no role in study design, data collection and interpretation, or the decision to submit the work for publication.

### Author contributions

Sergio Plaza-Alonso, Conceptualization, Formal analysis, Validation, Investigation, Visualization, Writing – original draft, Writing – review and editing; Nicolas Cano-Astorga, Validation, Investigation, Writing – review and editing; Javier DeFelipe, Conceptualization, Resources, Supervision, Funding

acquisition, Validation, Visualization, Writing – review and editing; Lidia Alonso-Nanclares, Conceptualization, Data curation, Supervision, Validation, Investigation, Visualization, Methodology, Project administration, Writing – review and editing

### Author ORCIDs
Sergio Plaza-Alonso ⓘ https://orcid.org/0000-0002-2484-5791
Nicolas Cano-Astorga ⓘ https://orcid.org/0000-0003-3724-0481
Javier DeFelipe ⓘ https://orcid.org/0000-0001-5484-0660
Lidia Alonso-Nanclares ⓘ https://orcid.org/0000-0003-2649-7097

### Ethics

Human brain tissue was obtained from 3 autopsies with no recorded neurological or psychiatric alterations (supplied by Unidad Asociada Neuromax, Laboratorio de Neuroanatomía Humana, Facultad de Medicina, Universidad de Castilla-La Mancha, Albacete, Spain). The consent of the individuals was obtained and the sampling procedure was approved by the Institutional Ethical Committee of the Albacete University Hospital. The tissue was obtained following national laws and international ethical and technical guidelines on the use of human samples for biomedical research purposes. Human brain tissue from the same subjects has been used in previous studies (e.g., Cano-Astorga et al., 2021, 2023; Dominguez-Alvaro et al., 2021a, 2021b; Montero-Crespo et al., 2020).

### Decision letter and Author response
Decision letter https://doi.org/10.7554/eLife.96144.sa1
Author response https://doi.org/10.7554/eLife.96144.sa2

---

# Additional files

### Supplementary files
Supplementary file 1. Supplementary tables. (a) Light microscopy data: volume fraction occupied by cortical elements in layers I, II-is, II-ni, III, Va/b, Vc, and VI of the MEC. Vc: volume fraction occupied by cells bodies; Vbv: volume fraction occupied by blood vessels; Vn: volume fraction occupied by neuropil. (b) Light microscopy data: volume fraction occupied by cortical elements in layers I, II-is, II-ni, III, Va/b, Vc, and VI of the MEC, for individual cases. Vc: volume fraction occupied by cells bodies; Vbv: volume fraction occupied by blood vessels; Vn: volume fraction occupied by neuropil. (c) Summary of the stack details obtained from the multiple sampling from all layers of MEC. Three stacks of images were acquired in each MEC layer, per case (nine stacks of images in total per layer). The last row shows the sum of all the MEC layers from all cases. (d) Accumulated data acquired from the ultrastructural analysis of neuropil from layers I, II-is, II-ni, III, Va/b, Vc, and VI of the MEC for individual cases. Data in parentheses are not corrected for shrinkage. AS: asymmetric synapses; CF: counting frame; SS: symmetric synapses. (e) Number of synaptic SAS analyzed (n), the location (μ), and scale (σ) of the best-fit log-normal distributions in the six cortical layers. AS: asymmetric synapses; SAS: synaptic apposition surface; SS: symmetric synapses. (f) Proportion of the different shapes of synaptic junctions in MEC layers. Data in parentheses refer to absolute numbers of synapses. (g) Proportion of the different shapes of synaptic junctions in MEC layers for individual cases. Data in parentheses refer to absolute numbers of synapses. (h) SAS area of asymmetric (AS) and symmetric (SS) synapses for each synaptic shape in each MEC layer. Data in parentheses are not corrected for shrinkage. Values (in $nm^2$) are expressed as mean ± SE. (i) SAS area of asymmetric (AS) and symmetric (SS) synapses for each synaptic shape, in each MEC layer, per individual case. Data in parentheses are not corrected for shrinkage. Values (in $nm^2$) are expressed as mean ± SE. (j) Proportion of the different postsynaptic targets in MEC layers. Synapses established on spine head include both complete and incomplete spines. Data in parentheses refer to the absolute number of synapses found in each layer. (k) Proportion of the postsynaptic targets of synaptic junctions in MEC layers, for individual cases. Total synapses on spine heads are calculated as the sum of 'Complete Spine Heads' and 'Incomplete Spine Heads' columns. Data in parentheses refer to the absolute numbers of synapses. (l) SAS area of asymmetric (AS) and symmetric (SS) synapses regarding the postsynaptic target in each MEC layer. Data in parentheses are not corrected for shrinkage. Values (in $nm^2$) are expressed as mean ± SE. (m) SAS area of asymmetric (AS) and symmetric (SS) synapses according to the postsynaptic target in each MEC layer, per individual case. Data in parentheses are not corrected for shrinkage. Values (in $nm^2$) are expressed as mean ± SE. (n) Coefficient of variation of the analyzed synaptic parameters in each MEC layer between the stack of images. AS: asymmetric

synapses; CV: coefficient of variation; SAS: synaptic apposition surface. (o) Data on postsynaptic targets in different species, regions, and cortical layers. EC: entorhinal cortex; F: female; FIB/SEM: focused ion beam-SEM; M: male; MEC: medial entorhinal cortex; TCE: transentorhinal Cortex; TEM: transmission electron microscopy. Age refers to years, except otherwise indicated. *No classification of the type of synapses (AS:SS) were performed. **Only axospinous synapses (established on dendritic spine heads and necks) and axodendritic synapses (formed on dendritic shafts) are indicated. ***No distinction between spiny and aspiny shafts were made. (p) Clinical and neuropsychological information from the cases analyzed.

Supplementary file 2. Spreadsheet containing the spatial extents of individual stacks of images, detailed per case, and layer. Original dimensions (x, y, z), original volumes, counting frame (CF) dimensions, and CF volume are provided.

Supplementary file 3. Spreadsheet containing Kruskal–Wallis and Dunn's tests p-values of the interindividual variability analysis, organized in TABS as follows. The criterion for statistical significance was considered to be met if p<0.05 (indicated in the spreadsheet as *) or p<0.01 (**).

> SYNAPTIC DENSITY
> INTERSINAPTIC DISTANCE
> SAS AREA OF AS
> SAS AREA OF SS
> SAS AREA OF MACULAR AS
> SAS AREA OF MACULAR SS
> SAS AREA OF COMPLEX-SHAPED AS
> SAS AREA OF COMPLEX-SHAPED SS

Supplementary file 4. Spreadsheet containing the complete raw dataset organized in TABS as follows: RAW DATA SYNAPSES: each row corresponds to each analyzed synapse and contains its characteristics, SAS area corrected for tissue shrinkage and SAS area provided by EspINA. RAW DATA DENSITIES: each row corresponds to each analyzed stack of images and contains the number of synapses per type, the volume of the counting frame, the volume of the counting frame corrected for tissue shrinkage and artifacts, and the calculated synaptic density per type and total. SIZE: mean SAS area, SE, and number of synapses calculated per stack of images, per synaptic type and layer. SHAPE: number of synapses, mean SAS area and SE per synaptic shape, calculated per stack of images, synaptic type, and layer. POSTSYNAPTIC: number of synapses, mean SAS area and SE per postsynaptic target, calculated per stack of images, synaptic type, and layer.

MDAR checklist

## Data availability

Most data are available in the main text or the supplementary files. Raw data used for analysis are provided in *Supplementary file 4*. Stack of images are available at BossDB. The datasets used and analyzed during the current study are published in the EBRAINS Knowledge Graph: https://doi.org/10.25493/3GMJ-FEZ, https://doi.org/10.25493/QGH8-MTS, https://doi.org/10.25493/CSB1-Q07.

The following datasets were generated:

| Author(s) | Year | Dataset title | Dataset URL | Database and Identifier |
|---|---|---|---|---|
| Plaza-Alonso S, Cano-Astorga N, DeFelipe J, Alonso-Nanclares L | 2024 | Volume electron microscopy to analyze the microanatomical features of synapses in all layers of the Medial Entorhinal Cortex from the human brain | https://doi.org/10.60533/boss-2024-ekc7 | BossDB, 10.60533/boss-2024-ekc7 |
| Alonso-Nanclares L, Cano-Astorga N, Plaza-Alonso S, DeFelipe J | 2022 | 3D ultrastructural study of synapses using FIB/SEM of deep layers from the human Entorhinal Cortex (v1) | https://doi.org/10.25493/QGH8-MTS | EBRAINS, 10.25493/QGH8-MTS |

*Continued on next page*

*Continued*

| Author(s) | Year | Dataset title | Dataset URL | Database and Identifier |
|---|---|---|---|---|
| Plaza-Alonso S, Cano-Astorga N, DeFelipe J, Alonso-Nanclares L | 2024 | 3D ultrastructural study of synapses using FIB/SEM of upper layers from the human Entorhinal Cortex (v1) | https://doi.org/10.25493/CSB1-Q07 | EBRAINS, 10.25493/CSB1-Q07 |

The following previously published dataset was used:

| Author(s) | Year | Dataset title | Dataset URL | Database and Identifier |
|---|---|---|---|---|
| Alonso-Nanclares L, Dominguez-Alvaro M, Blazquez-Llorca L, Montero-Crespo M, DeFelipe J | 2020 | 3D ultrastructural study of synapses using FIB/SEM in the human Entorhinal Cortex (v1) | https://doi.org/10.25493/3GMJ-FEZ | EBRAINS, 10.25493/3GMJ-FEZ |

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
