## [Editor Report]

This study presents a useful examination of dense neuroanatomy in human postmortem medial entorhinal cortex, using a large number of small electron microscopy image volumes sampled from multiple cortical layers and individuals. The authors use solid experimental and annotation techniques, demonstrating the suitability of postmortem tissue reconstructions for analysis and presenting careful, detailed measurements of synapse properties and overall tissue composition in this brain region. This work would be of interest for studies of cellular neuroanatomy or brain network organization.

---

## [Decision Letter]

**Decision letter after peer review:**

Thank you for submitting your article "Volume Electron Microscopy Reveals Unique Laminar Synaptic Characteristics in the Human Entorhinal Cortex" for consideration by *eLife*. Your article has been reviewed by 2 peer reviewers, including Alyssa Wilson as the Reviewing Editor and Reviewer #1, and the evaluation has been overseen by Albert Cardona as the Senior Editor. The following individual involved in the review of your submission has agreed to reveal their identity: Casey Schneider-Mizell (Reviewer #2).

Essential Revisions (for the authors):

In the reviewers' discussions, several claims from this article were identified that require further support or alteration. Of the reviewers' recommendations, the following revisions will be expected in follow-up work:

1) Throughout the manuscript, claims of homogeneity should be contextualized more fully, with discussions about the potential contributions of noise to observations, and about the applicability of findings given the spatial scales investigated and the measurements made. In this expanded discussion, please address the following points in particular:

(1a) For each ultrastructural measurement presented, a quantitative characterization of variability should be reported, on a per-image-volume basis (currently, measures appear to be aggregated per individual, per layer, as in Table S3). Additionally, a figure formatted similarly to Figure 3C but with variability measures (e.g. error bars with interquartile range per data point) should be presented for each measurement, to provide a clearer sense of inter-image-volume variability. The degree of this variability should be discussed in the text.

(1b) Similarly, given the substantial clinical differences between brain donors, a characterization of inter-individual variability should be presented, where distributions for each measurement are aggregated by donor, per layer, and compared (again using plots similar to Figure 3C but with variability measures). Here, statistical tests of similarity between donors would also be of interest for assessing the impacts of combining data for multiple donors.

(1c) In the text, the approximate size of each reconstructed volume (~10 µm per side) should be explicitly reported and discussed as it relates to the spatial scales for which structural variability can be interrogated. It would further be helpful to provide context about how these volumes may relate to structure at other spatial scales relevant for human medial entorhinal cortex.

2) Although data acquisition and annotation have been carefully executed in this work, there are several aspects of synapse selection and annotation that should be more thoroughly described, since these factors may have impacted the authors' findings. In particular;

(2a) The authors appear to annotate active zones and postsynaptic densities by thresholding synapse images at some user-defined pixel intensity value, taking only pixels darker than that threshold as their annotations (Lines 806 – 812). This technique seems like it could be prone to producing noisy annotations, particularly since in the EM images provided (Figures S11-16) the pixel intensities of active zones/postsynaptic densities and surrounding neuropil do not appear to be highly distinct.

(2b) The authors note that they have excluded synapses formed onto cell somata or proximal apical dendrites in their analyses. However, this choice may have resulted in disproportionate exclusion of inhibitory synapses, as inhibitory neurons are more likely to form synapses directly onto somata or dendritic shafts. The authors should quantify the number of synapses that were excluded as a result of this approach and should comment about potential impacts on their findings (for example the fraction of asymmetric synapses observed to form on dendritic shafts of spiny neurons as shown in Figure 6).

3) Much of the value of this study derives from the data itself, and further, in order to fully reproduce the findings in this work, the image volumes, corresponding synapse segmentations, and metadata should be made readily available (pending any agreements that may be needed for sharing human tissue data). Currently, it appears that segmentations and annotations, but not the image volumes, have been made available at the EBRAINS Knowledge Graph. Further, while not essential, it would substantially increase the usability of this dataset if it were deposited in a form compatible with industry standard tools for electron microscopy, like Neuroglancer or cloud-volume, and the authors are encouraged to consider this option.

*Reviewer #1 (Recommendations for the authors):*

I would like to emphasize that overall, this work contains an impressive amount of rigorous analysis. The main concern I have is that the variability in the analysis, some of which I understand is unavoidable given the work involved in EM reconstructions and in identifying appropriate brain donors, should be more explicitly shown. The claim of homogeneity should be stated in such a way that someone who reads the abstract, for instance, does not take away this piece of your findings without also understanding the important caveats.

*Reviewer #2 (Recommendations for the authors):*

Specific Recommendations and Concerns

1) The concept of "synaptic organization," as introduced on Line 122, is a key part of the framing but it is not clear what specifically the authors aim to measure. Some meanings could indeed be a "crucial aspect" but some might not. This matters when making claims about views of the "best strategy." There are many aspects to the synaptic organization of the nervous system that require whole-cell scale reconstruction and are completely unavailable to small cutouts or, indeed, anything less than whole-brain datasets. There are other aspects of synaptic organization, such as the ones considered in this work, where smaller cutouts are sufficient, and it is thus possible to generate better sampling. It is misleading to suggest that the detailed goals of and the observations produced by circuit-scale connectomics and small-cutout volume EM are the same, but that is not to say that one is any better than the other. The useful descriptions would be better served if the authors argued concretely why these properties they measure are important.

2) The most impressive part of this study is the sampling across multiple individuals, which is often not feasible in larger-scale EM studies, and the most consistent result is the similarity across them. However, the choice of data representation is inconsistent with regards to the replicates within and between individuals, which dilutes the ability to assess these trends. For example, Figure 3C very nicely shows each sample from each individual, while Figure 1A appears to group samples but still shows each individual, while panels like Figure 1B and Figure 4B-E omit these distinctions entirely. A more consistent and complete visualization of the inter-sample and inter-individual data throughout would strengthen the arguments of the paper. Much of this is already done in the supplementary figures, but could be brought more fully into the results and discussion (for example, variability is largely not discussed in terms of synapse shape or size).

3) While synapses onto excitatory dendrites are generally straight-forward to classify into asymmetric and symmetric synapses, the postsynaptic densities of synapses onto inhibitory dendrites are not as clearly defined in my experience. Similarly, experience has shown that manual detection of small synapses can be quite difficult even after several passes. It would be useful to know how the authors approached and mitigated these specific problems in their tissue.

4) I was surprised by the relatively large value of asymmetric synapses onto dendritic shafts of spiny neurons (10-20%, in most cases 4-5 times the number of symmetric synapses onto spiny shafts), and the consistency across layers, as shown in Figure 6. It does not match my experience in mammalian cortex that excitatory inputs onto excitatory shafts significantly outnumber inhibitory inputs onto shafts. Is this a difference between mouse/rodent and human MEC?

5) Much of the value of this study derives from the data itself. It would be extremely beneficial for the authors to deposit the image volumes, synapse segmentations, and metadata online in a form that could be visualized in industry standard tools like Neuroglancer and read by software such as cloud-volume. This could be especially nice if integrated into the tissue coordinates identified from the light level analysis to create a multiscale dataset. I do see the synapse segmentations on the EBRAINS Knowledge Graph platform, but not the associated imagery. Moreover, the.seg format is for Espina, which appears to not be available for Mac OS. Given that the authors are engaging on a series of similar studies.

[Editors' note: further revisions were suggested prior to acceptance, as described below.]

Thank you for resubmitting your work entitled "Volume Electron Microscopy Reveals Unique Laminar Synaptic Characteristics in the Human Entorhinal Cortex" for further consideration by *eLife*. Your revised article has been evaluated by Albert Cardona (Senior Editor), a Reviewing Editor, and a peer reviewer. Their feedback is below.

We thank the authors for the improvements made to the new manuscript, which have addressed many of our comments. We note that the authors have clarified the Results text, added some statistical test data, satisfactorily clarified their synapse annotation methods, and reformatted their plots to meet most of the Essential Revisions stipulations.

However, several points in the Essential Revisions still need to be addressed, as outlined below:

(1) As stipulated in Essential Revision 3, the authors are required to include a way to access their EM data in their Data Availability statement. Although we understand the authors' concerns, sharing raw data is journal policy.

(2) As stipulated in Essential Revision 1, the authors are required to present an explicit quantification of variability for each synapse measurement, in two respects:

* Essential Revision 1a): between the 3 image stacks collected per donor+layer.

* Essential Revision 1b): between donors per layer.

We appreciate several revisions and responses made by the authors with respect to these requirements (below). We would like to clarify what additional changes are required to satisfy Essential Revisions 1a) and 1b) beyond these updates.

– The authors have included a of coefficient of variation for each synapse measure by layer (using aggregated donor/stack data).

– While the coefficient of variation is a useful quantification of the per-layer data assuming it is well-mixed, on its own it unfortunately does not address our main concerns of whether the data itself is well-mixed for each layer, across individual stacks per donor and across donors.

– Data visualizations now show inter-stack variability to a much greater extent, with most properties plotted in the format of Figure 3C (Figure 1C, 3A,C, 4C,E; Figure 1S1A, 4S3A,C).

- This represents a substantial improvement but does not include sufficient information to assess data mixing. The image stack means are provided without a measure of variance, but the combination of both mean and variance is necessary for visualizing cross-stack variability. Currently, error bars correspond to per-layer aggregated data, which is useful for the across-layer comparisons but does not address the issue of variability we raised.

– The authors have noted in their response (Essential Revisions 1c) that sampling of multiple small image stacks within a tissue can be sufficient to capture the distributions of synapse properties, citing Merchan-Perez et al. 2009 (which studied synapse numbers in the cerebral cortex).

– We agree that sampling can be a reasonable strategy for the measurement of anatomical synapse properties and that reconstruction of an entire tissue is not necessarily preferable in these cases. However, given that every donor, tissue type/sample, and synapse property is subject to different variability, it is essential to perform a quantitative analysis of the data itself to show that sampling is sufficient for the synapse properties measured. One piece of this analysis would be a demonstration that stacks are well-mixed, which we are requiring. An additional piece would be a power analysis.

With these points in mind, the following changes are required:

2A) Essential Revision 1a): in Figures1C, 3A,C, 4C,E 1S1A, and 4S3A,C, each data point must be given its own error bar to allow visual comparison of stacks (+/-standard deviation is fine). Additionally, statistical comparisons of the 3 image stacks per donor+layer should be explicitly reported for each synapse measure. A nonparametrical test that compares distributions, like the Mann-Whitney/Kruskal Wallis (used by the authors elsewhere in the manuscript) is appropriate and should be used. Test results (p-values) can be reported in a supplemental table; any cases in which stacks for a donor+layer significantly differ should be noted in the figure legends. Some kind of commentary on these results should also be provided in the text.

(2B) Essential Revision (1b): a new figure panel showing inter-donor variability is required. This new figure should have a panel for each of the following measures: synapse density; intersynaptic distance; SAS synapse area; AS synapse area; macular-shaped synapse area (for SAS and AS synapses separately); complex-shaped synapse area (SAS and AS synapses separately). In each panel the mean + standard deviation should be shown for each donor. Statistical testing should also be performed to check whether donor distributions are statistically distinguishable (Mann-Whitney/Kruskal-Wallis is also fine) and p-values should be reported in a supplemental table, with statistical differences between donors noted in the figure panel/legend. Please also briefly comment on these comparisons in the text.

Notably, this step will be quite useful to the field to be able to cite cases like this when handling questions of inter-individual variability.

(3) The term "synaptic organization" must be replaced throughout the manuscript with a term that clearly communicates the types of measurements that were analyzed, to avoid misinterpretation. One suggestion for an acceptable term would be "anatomical synapse features of neuropil". (Essential Revision 1)

(4) The Abstract, Introduction, and Discussion sections should be revised to clearly indicate that anatomical features of neuropil synapses are being studied, and to clearly summarize the paper's findings as being structural patterns that are either repeated across cortical layers based on their dataset or not. The term "homogeneity" should be avoided. (Essential Revision 1)

(5) Essential Revision 1c): In the interest of transparency, the authors are required to report the (x,y,z) sizes of individual blocks in the manuscript.

In the current revision, the authors' reporting on individual block size unfortunately remains less than transparent. They do report a range and mean of individual block volumes in the text (Lines 212-213: "The number of sections per stack ranged from 244 to 319 (Supplementary file 1c), which corresponds to a raw volume ranging from 384 to 502 μm^3 (mean: 450 μm^3)."), but further clarification about what kind of spatial scale these blocks represent is not presented. Other information about block size is also indirect (inclusion volume per block is given in the "Raw Data Densities" sheet of Supplementary file 2; in Figure 9D FIBSEM trenches are shown at low magnification, with the relevant scale bar in a separate panel such that length estimation is difficult).

Being explicit about the 1-dimensional spatial extents of individual image stacks is important for the claims the authors are making. For example, as noted by Reviewer #2 previously, the author's use of F, G, and K functions only allow them to measure deviations in their data from random point distributions if the densities are changing at a scale smaller than their volume samples.

[Editors' note: further revisions were suggested prior to acceptance, as described below.]

Thank you for resubmitting your work entitled "Volume Electron Microscopy Reveals Unique Laminar Synaptic Characteristics in the Human Entorhinal Cortex" for further consideration by *eLife*. Your revised article has been evaluated by Albert Cardona (Senior Editor) and a Reviewing Editor.

The manuscript has been improved but there are some remaining issues that need to be addressed, as outlined below:

1. We greatly appreciate that the authors have shared the BOSSDB URL for their EM images. We would just ask that the DOI be added to the manuscript prior to acceptance.

2. The p-values for the Kruskal-Wallace (KW) tests used in assessing inter-individual variability should be provided in addition to the p-values reported for the Dunn's multiple comparison tests. We note that the Dunn test is a follow-up to the KW, and in this case would be used to investigate differences between donor pairs when the KW test indicates significant differences among the 3 donors. Unfortunately, Dunn's test can fail to detect differences for several reasons (it applies a z-score approximation to each donor distribution, and it can use conservative p-value corrections). For this reason, Dunn test p-values should be considered alongside their corresponding KW test p-value.

3. The Discussion section should be condensed substantially (one notable part is Lines 516-568), as this version is still difficult to read through.

4. Line 571: it is stated that synapses "were randomly distributed", but the statistical testing used only implies that no differences could be distinguished from random distributions, which is a different conclusion. Please adjust this phrasing to more accurately reflect the conclusions of these tests.

5. Please place a scale bar in each panel of Figure 9 separately, to help readers more easily interpret these images (currently the authors use one scale bar for all panels, and this makes it difficult to visualize sizes in some panels).

---

## [Author Response]

Essential Revisions (for the authors):In the reviewers' discussions, several claims from this article were identified that require further support or alteration. Of the reviewers' recommendations, the following revisions will be expected in follow-up work:1) Throughout the manuscript, claims of homogeneity should be contextualized more fully, with discussions about the potential contributions of noise to observations, and about the applicability of findings given the spatial scales investigated and the measurements made. In this expanded discussion, please address the following points in particular:1a) For each ultrastructural measurement presented, a quantitative characterization of variability should be reported, on a per-image-volume basis (currently, measures appear to be aggregated per individual, per layer, as in Table S3). Additionally, a figure formatted similarly to Figure 3C but with variability measures (e.g. error bars with interquartile range per data point) should be presented for each measurement, to provide a clearer sense of inter-image-volume variability. The degree of this variability should be discussed in the text.1b) Similarly, given the substantial clinical differences between brain donors, a characterization of inter-individual variability should be presented, where distributions for each measurement are aggregated by donor, per layer, and compared (again using plots similar to Figure 3C but with variability measures). Here, statistical tests of similarity between donors would also be of interest for assessing the impacts of combining data for multiple donors.

We agree that the present results could benefit from plotting and deeper examining the potential variability between individuals and image stacks for the analyzed parameters.

Accordingly, we have now modified the manuscript as follows:

We have adapted most of the plots to the same format used in Figure 3C, where each point represents the mean value obtained from a single stack of images from each individual and layer.

A new analysis to study the variability of each synaptic parameter has been performed. The coefficient of variation, both between cases and stack of images for each layer, has been calculated. New tables have been added (Tables 2 and Supplementary file 1n).

The following new subsection has been included in the Results section:

“Synaptic parameters show little variability

Variability between cases was examined by calculating the coefficient of variation (CV, see material and methods) of each synaptic parameter in every MEC layer.

We observed little variability between cases for any of the synaptic characteristics analyzed (Table 2). Overall, the SAS area of the AS exhibited the highest CV values of all the parameters analyzed, ranging from 3.3% in layer I to 19.6% in layer VI (see Table 2).

We also calculated the variability between the stack of images, examining the CV for the same synaptic parameter in each MEC layer. In this case, the highest variability was observed in the synaptic density in layer Vc (20.4%; Supplementary file 1n). However, the rest of the layers presented low variability regarding synaptic density. Furthermore, the SAS area of AS presented a relatively higher variability than the rest of the synaptic parameters analyzed, ranging from 10.2% in layer I to 19.4% in layer VI (Supplementary file 1n).”

-In the last subsection of the Discussion section, we have added the following paragraph:

“[…]. Indeed, the proportion of AS:SS showed remarkable uniformity across individuals in all layers (Table 1). This apparent similarity was further confirmed through the examination of the variability between cases (Table 2) and stacks of images (Supplementary file 1n), supporting the homogeneity of the data in all layers of the MEC. Furthermore, in all individuals, layers and stack of images, SS had significantly smaller sizes and were in much lower numbers than AS. This is in line with the fact that, as already discussed, synaptic characteristics of SS are similar in the different cerebral regions, layers and animals examined so far (e.g., Alonso-Nanclares et al., 2023; Cano-Astorga et al., 2023).

Overall, the present data point to a set of highly conserved synaptic characteristics in all layers of the MEC, while some data point to a laminar specificity. Future studies on human synaptic nanoconnectivity of different layers in additional brain regions could confirm whether there is a commonly shared trend across the human cortex.”

1c) In the text, the approximate size of each reconstructed volume (~10 µm per side) should be explicitly reported and discussed as it relates to the spatial scales for which structural variability can be interrogated. It would further be helpful to provide context about how these volumes may relate to structure at other spatial scales relevant for human medial entorhinal cortex.

In the present work, we aimed to investigate synaptic density, types, and sizes, including the identification of postsynaptic targets in all layers of the medial entorhinal cortex, a relatively large cortical region (approximately 1800 mm³). To accomplish this goal, our study involved a substantial sample, encompassing 63 FIB/SEM stacks of images, each consisting of 229 to 319 images, totaling around 18000 images, and a total examined volume of 28466 µm^3^.

We have added an additional table (Supplementary file 1c) detailing the sampling volume for each layer, including both the range and the total number of images per stack in each layer.

From our point of view, multiple sampling using volume electron microscopy is crucial when examining human brain tissue, not only because of the large size of the brain and extension of particular brain regions, but also because interindividual variability is more pronounced than in animal models. Thus, to explore the synaptic characteristics present in a given region, our first-choice approach is to determine the range of variability by sampling relatively small volumes of the region of interest multiple times and in several individuals.

Respectfully, we consider our sampling to be adequate to study the synaptic characteristics of the MEC neuropil and we believe that our study is of great significance. As far as we know, the results of this study constitute the largest synaptic dataset collected to date in the medial part of the human entorhinal cortex obtained from truly control brain samples.

Moreover, as discussed in Merchan-Perez et al. (2009), it is expected that the variation among the ultrastructural characteristics will also fall within narrow windows or, at least, the statistical distribution of that variation may be modeled. The estimation of this distribution can be achieved by means of spatial sampling strategies (e.g., Lafratta, 2006). We believe reconstructing the entire layer within a cortical region is unnecessary and impractical for defining the absolute synaptic number, type, and other ultrastructural characteristics. Multiple sampling of relatively small volumes within the region is sufficient to capture the range of variability.

Since our data showed remarkably little variability between layers (3 stacks of images per layer), we are confident that our sampling method accurately represents the reality of the synaptic characteristics within each layer of the MEC.

As we stated in our Response 1b, to better illustrate sampling variability, we have changed the plots to include the data from every stack of images from each individual per layer (similar to the format used for Figure 3C). In addition, we have performed an analysis of the coefficient of variation to examine the variability of the different stacks of images for each analyzed parameter.

2) Although data acquisition and annotation have been carefully executed in this work, there are several aspects of synapse selection and annotation that should be more thoroughly described, since these factors may have impacted the authors' findings. In particular;2a) The authors appear to annotate active zones and postsynaptic densities by thresholding synapse images at some user-defined pixel intensity value, taking only pixels darker than that threshold as their annotations (Lines 806 – 812). This technique seems like it could be prone to producing noisy annotations, particularly since in the EM images provided (Figures S11-16) the pixel intensities of active zones/postsynaptic densities and surrounding neuropil do not appear to be highly distinct.

It is important to note that the applied software in this study, EspINA, does not automatically identify any structure, it is a supervised software tool that performs the automated segmentation of synapses identified by experts. The identification of synapses was manually performed by an expert (in case of doubt, by the consensus of several experts). Then, the segmentation of each synapse was automatically performed by EspINA based on grayscale intensity. Each synapse segmentation was revised and validated by the user. In addition, synapse segmentations can be manually edited if necessary to accurately delimitate the active zone and postsynaptic density of each synapse. In other words, EspINA significantly accelerates 3D reconstruction, allowing users to supervise the process, adjust parameters, and validate the results.

To confirm the identity of inhibitory synapses (symmetric synapses), we traced their axons through the image stack and examined each synapse they formed. 3D reconstruction facilitated this process by allowing navigation and examination of both pre- and postsynaptic elements for each synapse.

To be clearer on this matter, we have changed the following paragraph in material and methods. It now reads:

“[…] EspINA software allows the user to supervise the result of segmentation and reconstruction; each synapse segmentation is revised and validated by the user, and the segmentation can be manually edited if necessary to accurately delimitate the active zone and postsynaptic density of each synapse.”

Please see the new Figure 12 to illustrate synapse segmentation, 3D reconstruction and extraction of the synaptic apposition surface.

2b) The authors note that they have excluded synapses formed onto cell somata or proximal apical dendrites in their analyses. However, this choice may have resulted in disproportionate exclusion of inhibitory synapses, as inhibitory neurons are more likely to form synapses directly onto somata or dendritic shafts. The authors should quantify the number of synapses that were excluded as a result of this approach and should comment about potential impacts on their findings (for example the fraction of asymmetric synapses observed to form on dendritic shafts of spiny neurons as shown in Figure 6).

We would like to point out that in the present study, we focused on the examination of the neuropil. The neuropil is where the vast majority of synapses are found (90–98%; Alonso-Nanclares et al., 2008; DeFelipe, 1999). It has been described that the neuropil comprises approximately 90% of the grey matter of the human temporal neocortex (Cano-Astorga et al., 2023; Shapson-Coe et al., 2024) and the present results in MEC showed similar values (Supplementary file 1a). Accordingly, the exclusion of the somata and proximal segments of axon and dendrites might comprise a maximum of around 10% of the total synapses. So, the present data on the number of SS can be interpreted as a good estimation of putative GABAergic synaptic contacts in the neuropil, but not in the perisomatic region.

While understanding axo-somatic and axo-axonic innervation is crucial for inhibitory control in cortical circuits, our FIB/SEM image stacks lack information on neuronal somata and proximal dendrites, thus limiting our ability to analyze the synaptic contribution specifically in neuronal somata. In this regard, our FIB/SEM stacks might include axo-axonic synapses by chance, but due to their rarity, random sampling would not be ideal for accurate estimates. Studying axo-somatic synapses would require a different approach. Since cell somata are large structures, especially compared to the high magnification needed for synapse identification with EM, a single image might only capture a portion of the soma, potentially missing some axo-somatic synapses. Techniques like light microscopy with specific markers could be useful for analyzing these connections (see Ostos et al., 2023. Quantitative analysis of the GABAergic innervation of the soma and axon initial segment of pyramidal cells in the human and mouse neocortex. Cereb Cortex 33(7):3882-3909. doi: 10.1093/cercor/bhac314). Indeed, to analyze the axon initial segments, we must obtain FIB/SEM stacks just below the soma of pyramidal cells (where the axon initial segment originates), whereas for the analysis of the soma, we have to reconstruct the membrane surface of the soma of neurons, by selecting this particular region of the neuron.

We would like to point out that for the present study, it took over three years to obtain and analyze the data to prepare this article. This gives an indication of just how time-consuming this type of research can be and highlights why it is necessary to focus on specific objectives and avoid widening the scope of the study excessively. Hopefully, segmented images of the temporal human neocortex released in Neuroglancer by Shapson-Coe et al. (2024) could be examined to address this issue.

We have added the following paragraph to the Discussion section:

“[…] The reason for this is that our FIB/SEM stacks of images may include axo-axonic synapses by chance, but, since these synapses are very infrequent, random sampling of FIB/SEM stacks in the neuropil would not be the most appropriate strategy. Investigating perisomatic innervation would undoubtedly enhance our comprehension of the synaptic organization of these cortical circuits. The study of the axo-somatic synapses on neuronal somata would require a different technical approach. Since neuronal somata are large structures, particularly in relation to the high magnification of EM necessary to identify the synapses, a soma may occupy a significant part (if not all) of any given EM image and there would clearly be very few —if any— synapses to be analyzed. Thus, to analyze both axo-axonic and axo-somatic synapses, different technical approaches would need to be applied (for example, see Ostos et al., 2023).”

3) Much of the value of this study derives from the data itself, and further, in order to fully reproduce the findings in this work, the image volumes, corresponding synapse segmentations, and metadata should be made readily available (pending any agreements that may be needed for sharing human tissue data). Currently, it appears that segmentations and annotations, but not the image volumes, have been made available at the EBRAINS Knowledge Graph. Further, while not essential, it would substantially increase the usability of this dataset if it were deposited in a form compatible with industry standard tools for electron microscopy, like Neuroglancer or cloud-volume, and the authors are encouraged to consider this option.

We have now included the complete raw dataset obtained from our FIB/SEM images which was used to calculate all the measurements and analysis presented as Supplementary file 2. Since we are still working on the examination of some subcellular elements to be further analyzed, a deposit of the original stacks of images is not yet available.

Reviewer #2 (Recommendations for the authors):Specific Recommendations and Concerns1) The concept of "synaptic organization," as introduced on Line 122, is a key part of the framing but it is not clear what specifically the authors aim to measure. Some meanings could indeed be a "crucial aspect" but some might not. This matters when making claims about views of the "best strategy." There are many aspects to the synaptic organization of the nervous system that require whole-cell scale reconstruction and are completely unavailable to small cutouts or, indeed, anything less than whole-brain datasets. There are other aspects of synaptic organization, such as the ones considered in this work, where smaller cutouts are sufficient, and it is thus possible to generate better sampling. It is misleading to suggest that the detailed goals of and the observations produced by circuit-scale connectomics and small-cutout volume EM are the same, but that is not to say that one is any better than the other. The useful descriptions would be better served if the authors argued concretely why these properties they measure are important.

We agree with the reviewer that there are many aspects to the synaptic organization that require whole-cell scale analysis. We are sorry we were not clear enough using the term “synaptic organization”. We have now added the following definition to the Introduction section:

“We can distinguish two major goals in studying synaptic organization. First, studying connections between identified neurons, which consists of identifying the specific presynaptic and postsynaptic neurons involved in each synapse. Second, studying synaptic features in general, which involves quantifying synaptic density, identifying different types and sizes of synapses, and determining their postsynaptic targets (dendrites, somas, or other structures). In the present study, we focused on the second goal, but for simplicity, we refer to it as "synaptic organization" throughout the text.”

Regarding the importance of the synaptic properties examined in the present work, we consider that the provided dataset is useful not only for understanding local connectivity, but also for defining the MEC from a functional perspective.

To point out the importance of the synaptic properties analyzed, the following paragraphs have been added to the Introduction and Discussion sections:

“[…] Assessing the synaptic density in a particular brain region is crucial for understanding both connectivity and functionality. In particular, both the synaptic density (number of synapses per volume) and excitatory/inhibitory proportions are meaningful parameters to describe the synaptic connections of a particular brain region, which can be considered useful for understanding the synaptic circuits of any brain region.”

“Determination of the individual synaptic characteristics (such as synaptic size and shape) provides crucial data on the functionality of the synapses, regarding both synaptic transmission efficiency and dynamics. It has been proposed that synaptic size correlates with release probability, synaptic strength, efficacy, and plasticity (see Chindemi et al., 2022 and references therein). It has been demonstrated that synaptic size correlates with the number of receptors in the PSD; larger PSDs have more receptors (as reviewed in Lüscher et al., 2000; Lüscher and Malenka, 2012; Toni et al., 2001; Magee and Grienberger, 2020; Sumi and Harada, 2020).

[…] However, it has been reported that some receptors can be found both synaptically and extra-synaptically (Palomero-Gallagher and Zilles, 2019). Therefore, caution should be exercised when considering potential correlations between receptor densities, number of synapses and/or synaptic size.”

“Examining the postsynaptic target for each 3D reconstructed synapse may allow us to identify variations in the proportion of synapses on dendritic spines and shafts. These differences might signify a microanatomical specialization of the examined cortical layers. Such variations could have significant functional implications for information processing. Studies have shown (e.g., Cornejo et al., 2022) that the membrane potential of the postsynaptic neuron is modulated differently depending on whether a synapse is established on a dendritic spine or a dendritic shaft. Consequently, variations in the proportions of these synaptic targets have clear functional significance.”

2) The most impressive part of this study is the sampling across multiple individuals, which is often not feasible in larger-scale EM studies, and the most consistent result is the similarity across them. However, the choice of data representation is inconsistent with regards to the replicates within and between individuals, which dilutes the ability to assess these trends. For example, Figure 3C very nicely shows each sample from each individual, while Figure 1A appears to group samples but still shows each individual, while panels like Figure 1B and Figure 4B-E omit these distinctions entirely. A more consistent and complete visualization of the inter-sample and inter-individual data throughout would strengthen the arguments of the paper. Much of this is already done in the supplementary figures, but could be brought more fully into the results and discussion (for example, variability is largely not discussed in terms of synapse shape or size).

We have followed the reviewer’s advice. Please seerResponses to 1a and 1b to the Essential revisions.

3) While synapses onto excitatory dendrites are generally straight-forward to classify into asymmetric and symmetric synapses, the postsynaptic densities of synapses onto inhibitory dendrites are not as clearly defined in my experience. Similarly, experience has shown that manual detection of small synapses can be quite difficult even after several passes. It would be useful to know how the authors approached and mitigated these specific problems in their tissue.

Please see response to 2a to Essential revisions.

4) I was surprised by the relatively large value of asymmetric synapses onto dendritic shafts of spiny neurons (10-20%, in most cases 4-5 times the number of symmetric synapses onto spiny shafts), and the consistency across layers, as shown in Figure 6. It does not match my experience in mammalian cortex that excitatory inputs onto excitatory shafts significantly outnumber inhibitory inputs onto shafts. Is this a difference between mouse/rodent and human MEC?

To our knowledge, there are no volume EM studies performed on the mouse EC. In a study performed in the mouse somatosensory cortex (S1HL region) using the same techniques (FIB/SEM) and considering only synapses established on spiny shafts, asymmetric synapses slightly outnumbered symmetric synapses: 10.25% for AS and 7.35% for SS (Turégano-López, 2022).

In the human, this ratio has been described to be more variable. For example, in layer III of associative (BA21; BA24 and BA38) and primary cortex (BA3b and BA4) of the human cortex, the percentage of AS on spiny shafts was around 10–14%, whereas the percentage of SS on spiny shafts was 2–3% (Cano-Astorga et al., 2024, biorxiv), which is in line with the results shown here for the MEC. In the human CA1 region of the hippocampus, while the *stratum oriens* exhibited a slightly higher number of AS than SS established on spiny shafts, in the rest of the CA1 layers, the numbers of AS and SS on spiny shafts were quite similar, if not higher for SS (Montero-Crespo et al., 2022).

In fact, quantitative analyses of synapses in the neuropil of different mammals, including human, have shown that most synapses on dendritic shafts are AS (∼80%), with relatively few being SS (∼20%; Beaulieu et al., 1992; Peters et al., 2008; Hsu et al., 2017; Calì et al., 2018; Santuy et al., 2018b; Domínguez-Álvaro et al., 2019, 2021a,b; Yakoubi et al., 2019b; Montero-Crespo et al., 2020, 2021; Cano-Astorga et al., 2021, 2023; Alonso-Nanclares et al., 2023). This is in line with the results presented here, where the vast majority of synapses on shafts were AS (87%, please see Figure 6—figure supplement 1, panel B).

To be clearer on this matter, we have added an additional table showing data on the distribution of synapses established on the different postsynaptic targets from a variety of mammal species and brain regions (Supplementary file 1o).

Additionally, we have modified Figure 6 to include the proportion of AS established either on dendritic spines or shafts, as well as the proportion of SS on dendritic spines or shafts, for each MEC layer. Accordingly, we have also included the following in the Results section:

“We also analyzed the proportion of excitatory (AS) and inhibitory (SS) synapses separately, categorized by their location on dendritic spines (heads or necks) or dendritic shafts. The proportion of AS formed on the different postsynaptic targets greatly resembled the general distribution of synapses established on the postsynaptic targets described above, given the greater number of AS than SS. In all layers of the MEC, AS were mostly established on dendritic spine heads (63%), followed by AS established on dendritic shafts (36.5%; 17.6% on spiny shafts and 18.9% on aspiny shafts) and, lastly, we found a small percentage of AS formed on dendritic spine necks (0.6%; Supplementary file 1k). Our analysis revealed significant differences between MEC layers regarding the proportions of AS synapses established on dendritic spine heads. Layer VI exhibited the highest proportion of AS on spine heads (80%), whereas layer Va/b had the lowest proportion (55%; χ2, p<0.0001). These findings are illustrated in Figure 6 and Supplementary file 1k.

The distribution of SS on different postsynaptic targets differed significantly from that observed for AS. The majority of SS were found on dendritic shafts (85.7%), with 47.6% on shafts containing dendritic protrusions (spiny shafts) and 38.1% on smooth dendritic shafts (aspiny shafts). A smaller proportion of SS were established on dendritic spine heads (12.1%) and spine necks (2.2%). Due to the low number of SS identified as axospinous in each layer (ranging from 3 in layer Va/b to 17 in layer Vc), statistical comparisons of this proportion between layers were not performed. However, we observed some variations between layers. Layer Va/b had the highest proportion of SS on dendritic shafts (92.3%), whereas layer Vc exhibited the lowest percentage of SS formed on shafts (77.4%; Figure 6; Supplementary file 1k).”

5) Much of the value of this study derives from the data itself. It would be extremely beneficial for the authors to deposit the image volumes, synapse segmentations, and metadata online in a form that could be visualized in industry standard tools like Neuroglancer and read by software such as cloud-volume. This could be especially nice if integrated into the tissue coordinates identified from the light level analysis to create a multiscale dataset. I do see the synapse segmentations on the EBRAINS Knowledge Graph platform, but not the associated imagery. Moreover, the.seg format is for Espina, which appears to not be available for Mac OS. Given that the authors are engaging on a series of similar studies.

To address this point, we have now included the complete raw dataset obtained from our FIB/SEM images which was used to calculate all the measurements and analysis, presented in the manuscript as Supplementary file 2.

[Editors’ note: what follows is the authors’ response to the second round of review.]

We thank the authors for the improvements made to the new manuscript, which have addressed many of our comments. We note that the authors have clarified the Results text, added some statistical test data, satisfactorily clarified their synapse annotation methods, and reformatted their plots to meet most of the Essential Revisions stipulations.However, several points in the Essential Revisions still need to be addressed, as outlined below:(1) As stipulated in Essential Revision 3, the authors are required to include a way to access their EM data in their Data Availability statement. Although we understand the authors' concerns, sharing raw data is journal policy.

We understand the importance of data sharing and have made significant efforts to comply with the journal's policy. As requested, we have included the complete raw data used for our analysis as Supplementary File 4. Additionally, the derived datasets generated by our software analysis are accessible through links to the EBRAINS repository, as detailed in the Data and Materials Availability section. We believe these data provide sufficient information to verify our findings and interpretations.

If the journal specifically requires the EM image stacks, we would like to clarify that while committed to data transparency, we are currently unable to share the complete EM image stacks due to ongoing segmentation and analysis processes. Premature release of these stacks could compromise these efforts. Extracting valuable information from these stacks is a time-consuming process that requires significant human resources, which are currently limited in our laboratory. However, we are exploring options to make these stacks accessible upon specific request, while still protecting our ongoing research.

(2) As stipulated in Essential Revision 1, the authors are required to present an explicit quantification of variability for each synapse measurement, in two respects:* Essential Revision 1a): between the 3 image stacks collected per donor+layer.* Essential Revision 1b): between donors per layer.We appreciate several revisions and responses made by the authors with respect to these requirements (below). We would like to clarify what additional changes are required to satisfy Essential Revisions 1a) and 1b) beyond these updates.– The authors have included a of coefficient of variation for each synapse measure by layer (using aggregated donor/stack data).– While the coefficient of variation is a useful quantification of the per-layer data assuming it is well-mixed, on its own it unfortunately does not address our main concerns of whether the data itself is well-mixed for each layer, across individual stacks per donor and across donors.– Data visualizations now show inter-stack variability to a much greater extent, with most properties plotted in the format of Figure 3C (Figure 1C, 3A,C, 4C,E; Figure 1S1A, 4S3A,C).- This represents a substantial improvement but does not include sufficient information to assess data mixing. The image stack means are provided without a measure of variance, but the combination of both mean and variance is necessary for visualizing cross-stack variability. Currently, error bars correspond to per-layer aggregated data, which is useful for the across-layer comparisons but does not address the issue of variability we raised.– The authors have noted in their response (Essential Revisions 1c) that sampling of multiple small image stacks within a tissue can be sufficient to capture the distributions of synapse properties, citing Merchan-Perez et al. 2009 (which studied synapse numbers in the cerebral cortex).– We agree that sampling can be a reasonable strategy for the measurement of anatomical synapse properties and that reconstruction of an entire tissue is not necessarily preferable in these cases. However, given that every donor, tissue type/sample, and synapse property is subject to different variability, it is essential to perform a quantitative analysis of the data itself to show that sampling is sufficient for the synapse properties measured. One piece of this analysis would be a demonstration that stacks are well-mixed, which we are requiring. An additional piece would be a power analysis.With these points in mind, the following changes are required:2A) Essential Revision 1a): in Figures1C, 3A,C, 4C,E 1S1A, and 4S3A,C, each data point must be given its own error bar to allow visual comparison of stacks (+/-standard deviation is fine). Additionally, statistical comparisons of the 3 image stacks per donor+layer should be explicitly reported for each synapse measure. A nonparametrical test that compares distributions, like the Mann-Whitney/Kruskal Wallis (used by the authors elsewhere in the manuscript) is appropriate and should be used. Test results (p-values) can be reported in a supplemental table; any cases in which stacks for a donor+layer significantly differ should be noted in the figure legends. Some kind of commentary on these results should also be provided in the text.

We have consulted experts on statistical analysis (Unidad de Tecnologías Ómicas at Instituto Cajal) regarding this issue. We appreciate the reviewer's suggestion to conduct statistical comparisons between the three image stacks per donor+layer for each synapse measure. However, synaptic density and intersynaptic distance are each represented by a single count (or ratio) per stack, making traditional statistical comparisons inappropriate [1]. While inter-donor variability was assessed, as detailed in response 2B, we acknowledge that non-parametrical tests to compare distributions of large samples (each stack of images contains hundreds of synapses) can easily detect as significant non-meaningful differences expected from minimal variability between biological samples [2,3].

Please note that for synapse size, our primary interest lies in comparing differences between donor-layers. Although variability between stacks within the same donor-layer exists, as is expected in biological samples, we believe that focusing on differences between donor-layers is more critical to addressing our research question. To further clarify this point, we have included new figures in the revised manuscript to illustrate inter-individual variability for all the synaptic parameters (please see response 2B).

References:

[1] Jiang, J. (2022) Large Sample Techniques for Statistics. Textbook, SpringerLink

[2] ForstMeier, W. et al. (2017) Detecting and avoiding likely false-positive findings – a practical guide. Biol Rev Camb Philos Soc. 2017 Nov;92(4):1941-1968. doi: 10.1111/brv.12315. Epub 2016 Nov 23. PMID: 27879038.

[3] Greenland, S. et al. (2016) Statistical tests, P values, confidence intervals, and power: a guide to misinterpretations. European Journal of Epidemiology, 31 (337-350). doi: 10.1007/s10654-016-0149-3

2B) Essential Revision 1b): a new figure panel showing inter-donor variability is required. This new figure should have a panel for each of the following measures: synapse density; intersynaptic distance; SAS synapse area; AS synapse area; macular-shaped synapse area (for SAS and AS synapses separately); complex-shaped synapse area (SAS and AS synapses separately). In each panel the mean + standard deviation should be shown for each donor. Statistical testing should also be performed to check whether donor distributions are statistically distinguishable (Mann-Whitney/Kruskal-Wallis is also fine) and p-values should be reported in a supplemental table, with statistical differences between donors noted in the figure panel/legend. Please also briefly comment on these comparisons in the text.Notably, this step will be quite useful to the field to be able to cite cases like this when handling questions of inter-individual variability.

To illustrate inter-individual variability, we have generated new plots for the following synaptic parameters: synapse density; intersynaptic distance; SAS area of AS and SS; SAS area of macular-shaped AS and SS (separately); SAS area of complex-shaped AS and SS (separately) per layer. These are now provided in the new additional figures: Figure 1—figure supplement 2; Figure 3—figure supplement 1; Figure 4—figure supplement 4 and 5.

The suggested statistical comparisons have been performed, and the P-values are included in an additional spreadsheet (Supplementary File 3).

The manuscript has also been modified to mention these comparisons, and changes have been made to figure legends accordingly. The subsection “Study of the sample variability” includes the new analyses.

(3) The term "synaptic organization" must be replaced throughout the manuscript with a term that clearly communicates the types of measurements that were analyzed, to avoid misinterpretation. One suggestion for an acceptable term would be "anatomical synapse features of neuropil". (Essential Revision 1)

As suggested by the reviewer, we have replaced the term “synaptic organization” throughout the manuscript with "ultrastructural synaptic features", “ultrastructural synaptic characteristics” or “synaptic features”, as appropriate. We believe these terms more accurately reflect the specific focus of our study on synaptic ultrastructure. We have retained the term “synaptic organization” only when referring to the broader concept encompassing multiple synaptic characteristics, such as cortical region connectivity, PSD composition, synaptic transmission strength, and others.

(4) The Abstract, Introduction, and Discussion sections should be revised to clearly indicate that anatomical features of neuropil synapses are being studied, and to clearly summarize the paper's findings as being structural patterns that are either repeated across cortical layers based on their dataset or not. The term "homogeneity" should be avoided. (Essential Revision 1)

According the reviewer’s suggestion, the manuscript has been modified to address these issues.

(5) Essential Revision 1c): In the interest of transparency, the authors are required to report the (x,y,z) sizes of individual blocks in the manuscript.

A new table with detailed information on the individual blocks is now provided in Supplementary File 2. This spreadsheet contains the spatial extents of the 63 individual stacks of images, detailed per case and layer. The original dimensions (x, y, z) and volumes of the stacks obtained by the FIB/SEM are included. Additionally, the dimensions and volume of the counting frame used for analyses are also provided.

In the current revision, the authors' reporting on individual block size unfortunately remains less than transparent. They do report a range and mean of individual block volumes in the text (Lines 212-213: "The number of sections per stack ranged from 244 to 319 (Supplementary file 1c), which corresponds to a raw volume ranging from 384 to 502 μm^3 (mean: 450 μm^3)."), but further clarification about what kind of spatial scale these blocks represent is not presented. Other information about block size is also indirect (inclusion volume per block is given in the "Raw Data Densities" sheet of Supplementary file 2; in Figure 9D FIBSEM trenches are shown at low magnification, with the relevant scale bar in a separate panel such that length estimation is difficult).Being explicit about the 1-dimensional spatial extents of individual image stacks is important for the claims the authors are making. For example, as noted by Reviewer #2 previously, the author's use of F, G, and K functions only allow them to measure deviations in their data from random point distributions if the densities are changing at a scale smaller than their volume samples.

[Editors’ note: what follows is the authors’ response to the third round of review.]

The manuscript has been improved but there are some remaining issues that need to be addressed, as outlined below:1. We greatly appreciate that the authors have shared the BOSSDB URL for their EM images. We would just ask that the DOI be added to the manuscript prior to acceptance.

We have added the DOI to the BossDB URL as requested.

2. The p-values for the Kruskal-Wallace (KW) tests used in assessing inter-individual variability should be provided in addition to the p-values reported for the Dunn's multiple comparison tests. We note that the Dunn test is a follow-up to the KW, and in this case would be used to investigate differences between donor pairs when the KW test indicates significant differences among the 3 donors. Unfortunately, Dunn's test can fail to detect differences for several reasons (it applies a z-score approximation to each donor distribution, and it can use conservative p-value corrections). For this reason, Dunn test p-values should be considered alongside their corresponding KW test p-value.

Following this recommendation, p-values for the KW tests have been added to the Supplementary Table 3. The final results remain unchanged, as no additional differences were detected in the Kruskal-Wallis test. To be clearer on this matter, we have changed the following paragraph in “Synaptic parameters show little variability” section. It now reads:

“Statistical analyses only showed significant differences in two particular comparisons: SAS area of AS in layer II-is (KW, p<0.05) and SAS area of complex AS in layer VI (KW, p<0.01); in both analysis, differences were found between AB2 and AB7 (Dunn’s test, p<0.05; Figure 3—figure supplement 1; Figure 4—figure supplement 5; Supplementary File 3).”

3. The Discussion section should be condensed substantially (one notable part is Lines 516-568), as this version is still difficult to read through.

Discussion has been shortened in the suggested section accordingly.

4. Line 571: it is stated that synapses "were randomly distributed", but the statistical testing used only implies that no differences could be distinguished from random distributions, which is a different conclusion. Please adjust this phrasing to more accurately reflect the conclusions of these tests.

This section has been rephrased as suggested. Thank you for the feedback.

5. Please place a scale bar in each panel of Figure 9 separately, to help readers more easily interpret these images (currently the authors use one scale bar for all panels, and this makes it difficult to visualize sizes in some panels).

A separate scale bar has been added to each panel in Figure 9 to facilitate the interpretation.